# OClO as observed by TROPOMI: a comparison with meteorological parameters and PSC observations

Jānis Puķīte[1], Christian Borger[1], Steffen Dörner[1], Myojeong Gu[1], and Thomas Wagner[1]

[1]Max Planck Institute for Chemistry, Mainz

**Correspondence:** Jānis Puķīte (janis.pukite@mpic.de)

**Abstract.** Chlorine dioxide (OClO) is a by-product of the ozone depleting halogen chemistry in the stratosphere. Although being rapidly photolysed at low solar zenith angles (SZAs) it plays an important role as an indicator of the chlorine activation in polar regions during polar winter and spring at twilight conditions because of the nearly linear dependence of its formation on chlorine oxide (ClO).

Here, we compare slant column densities (SCDs) of chlorine dioxide (OClO) retrieved by means of differential optical absorption spectroscopy (DOAS) from spectra measured by the TROPOspheric Monitoring Instrument (TROPOMI) with meteorological data for both Antarctic and Arctic regions for the first three winters in each of the hemispheres (November 2017 – October 2020). TROPOMI, a UV-VIS-NIR-SWIR instrument on board of the Sentinel-5P satellite monitors the Earth's atmosphere in a near polar orbit at an unprecedented spatial resolution and signal to noise ratio and provides daily global coverage

at the equator and thus even more frequent observations at polar regions.

The observed OClO SCDs are generally well correlated with the meteorological conditions in the polar winter stratosphere: e.g. the chlorine activation signal appears as a sharp gradient in the time series of the OClO SCDs once the temperature drops to values well below the Nitric Acid Trihydrate (NAT) existence temperature ($T_{NAT}$). Also a relation of enhanced OClO values at lee sides of mountains can be observed at the beginning of the winters indicating a possible effect of lee waves on chlorine

activation.

The dataset is also compared with CALIPSO Cloud-Aerosol Lidar with Orthogonal Polarization (CALIOP) polar stratospheric cloud (PSC) observations. In general, OClO SCDs coincide well with CALIOP measurements for which PSCs are detected.

Very high OClO levels are observed for the northern hemispheric winter 2019/2020 with an extraordinarly long period with

a stable polar vortex being even close to the values found for Southern Hemispheric winters. Also the extraordinary winter in 2019 in the Southern Hemisphere with a minor sudden stratospheric warming at the beginning of September was observed. In this winter similar OClO values were measured in comparison to the previous (usual) winter till that event but with a $1 - 2$ week earlier OClO deactivation.

# 1 Introduction

It is well established that catalytic halogen chemistry is responsible for stratospheric ozone depletion in polar regions in spring (WMO, 2018). The stratospheric dynamics are a key meteorological driving factor of chlorine activation: Towards winter the stratosphere above the poles cools down, leading to a strong meridional temperature gradient in the stratosphere. A balance between the temperature gradient and the vertical wind shear with strong westerly winds leads to the formation of the polar vortex (Lee, 2020). The Antarctic winters are generally characterized by a very stable polar vortex which is usually not the case

for Arctic winters. In this regard Lee (2020) summarizes that in the Arctic major stratospheric warmings (defined as easterly zonal mean winds at 10hPa and 60°N) take place every other winter while in the Antarctic such an event so far has been only observed in 2002. Once the air within the polar vortex cools down below a certain threshold (which varies with altitude), polar stratospheric clouds (PSCs) can form providing surfaces for the heterogeneous reactions of the chlorine activation (Solomon, 1999). In particular, $Cl_2$ is released in large amounts by the heterogeneous reaction of $ClONO_2$ and HCl. Once the air mass with

$Cl_2$ becomes irradiated by sunlight, $Cl_2$ is subsequently photolysed to atomic Cl (Solomon et al., 1986). Atomic Cl can result also from other reactions like between $ClONO_2$ and liquid or solid phase $H_2O$ and subsequent photolysis of the produced HOCl or other reactions (e.g. Nakajima et al., 2020). Atomic Cl in turn reacts with ozone (Stolarski and Cicerone, 1974). Because the resulting ClO (with or without involvement of BrO) is returned to atomic Cl (Molina and Molina, 1987; McElroy et al., 1986) by further reactions, a very effective ozone depletion process takes place. Furthermore, chlorine dioxide (OClO)

is a possible outcome of a reaction between ClO and BrO (Sander and Friedl, 1989):

$$ClO + BrO \rightarrow Br + OClO \tag{R1}$$

The dominant loss mechanism for atmospheric OClO is its very rapid photolysis (Solomon et al., 1990):

$$OClO + h\nu \rightarrow ClO + O \tag{R2}$$

which results in a null cycle with respect to ozone loss by recycling odd oxygen. Thus, OClO can be used as an indicator

for halogen chemistry because of the nearly linear dependence of OClO formation to ClO and BrO concentrations (Schiller and Wahner, 1996) at high solar zenith angles where the photolysis is slow enough to provide OClO abundances above the detection limit for passive scattered light UV/VIS measurements (Solomon et al., 1987).

PSCs are generally classified in three types: nitric acid trihydrate (NAT), supercooled ternary solution droplets (STS) and ice (e.g. Tritscher et al., 2021). There is an ongoing discussion about the forming temperatures and processes of the different

PSC components which in turn drive the temperature dependency of chlorine activation (Peter and Groß, 2012; Tritscher et al., 2021). While already formed NAT particles can exist below a certain temperature $T_{NAT}$, their formation pathway is supposed to be heterogeneous and is reported to start at about 3 K below this threshold (Peter et al., 1991; Koop et al., 1995; Voigt et al., 2005). Supercooled ternary solution droplets (STS) are formed at similar temperatures (around 3 K below $T_{NAT}$) (Carslaw et al., 1994). While occuring at similar rate per unit surface area density on different PSC type particles, it is attributed that the winter

chlorine activation is typically dominated by this (liquid) PSC type because of usually greater surface area density (Tritscher et al., 2021). Ice particles can form below the ice freezing temperature $T_{ICE}$ serving also as an additional condensation nuclei for the formation of mixtures for different PSCs types (Koop et al., 1995; Tritscher et al., 2021). It is worth mentioning that besides the chlorine activation on PSCs, a substantial onset in chlorine activation (already at temperatures around $T_{NAT}$) as caused by reactions on cold binary sulfate aerosol has been suggested (Drdla and Müller, 2012) but not without a controversy because

Solomon et al. (2015) have not found such a contribution.

Values of $T_{NAT}$ and $T_{ICE}$ are altitude dependent and there is also an impact of the atmospheric concentrations of their building species (Larsen, 2000). In our plots we consider $T_{NAT}$ and $T_{ICE}$ calculated for $HNO_3$ concentration of 8 ppbv and 5 ppmv for $H_2O$, representing typical winter conditions (Achtert et al., 2011, and references therein), and refer to $T'_{NAT} = T_{NAT} - 3$ K as the expected temperature for the PSC (i.e. NAT and STS) formation.

Chlorine starts to deactivate when PSCs evaporate (temperature rises above $T_{NAT}$) by converting most chlorine into the form of the reservoir species $ClONO_2$ with concentrations higher than before the activation (Müller et al., 1994). This deactivation process takes one to two weeks depending on the nitrate concentration (Kühl et al., 2004b). The time necessary for the deactivation is basically related to the time period and area with cold temperatures that existed beforehand and allowing PSC particle grow-up, which consequently can sediment faster for larger particles (Mann et al., 2003). Thus meanwhile ozone depletion

can continue even at temperatures above $T_{NAT}$ and chlorine activation can resume on a full scale once the air is cooled again and PSCs are reformed. Another possibility for chlorine deactivation is when almost complete destruction of ozone occurs and almost all chlorine becomes bound in HCl and cannot be reactivated even at cold temperatures because the necessary reaction partners $ClONO_2$ and HOCl are missing (Grooß et al., 2011). The conversion of the active chlorine into HCl can be quick: Grooß et al. (2011) reported timescales of $\sim$6h within their model run. This pathway can be found in the Antarctic where the

vortex is stable and cooling is persistently below $T_{NAT}$ for the whole winter and spring, however it can occur also for very cold stratospheric winters in the Arctic like it was the case for winter 2019/2020 (e.g. Manney et al., 2020; Grooß and Müller, 2021). As Nakajima et al. (2020) showed, the deactivation path can even depend on altitude.

For the first time OClO was measured by Solomon et al. (1987) by a ground based spectrograph in Antarctica contributing to a better undersanding of the extent in which the halogen chemistry is responsible for causing the recently discovered (Farman

et al., 1985) ozone hole. Shortly afterwards (Solomon et al., 1988) OClO abundances explainable only by heterogeneous chemistry were measured also for the Arctic. Several other studies for both polar regions followed (e.g. Kreher et al., 1995; Gil et al., 1996). Opportunities for global monitoring of OClO were enabled by satellite measurements when the GOME-1 instrument was launched in 1995 (Burrows et al., 1999). Many studies investigating the polar stratospheric chlorine activation were performed for GOME-1 OClO data (Wagner et al., 2001, 2002; Weber et al., 2002, 2003; Kühl et al., 2004a, b; Richter et

al., 2005). Later also measurements by SCIAMACHY, OSIRIS, OMI or GOME-2 were available for OClO analysis (Kühl et al., 2006; Krecl et al., 2006; Kühl et al., 2008; Puķīte et al., 2008; Oetjen et al., 2011; Hommel et al., 2014; Weber et al., 2021).

The TROPOspheric Monitoring Instrument (TROPOMI) is a UV-VIS-NIR-SWIR nadir viewing instrument on board of the Sentinel-5P satellite developed for monitoring the Earth's atmosphere (Veefkind et al., 2012). It was launched on 13 October 2017 in a near polar orbit and measures spectrally resolved earthshine radiances at an unprecedented spatial resolution of

around 3.5x7.2 km² (near nadir) at a high signal-to-noise ratio. It has a total swath width of ~2600 km on the Earth's surface providing daily global coverage (at equator) and a coverage of 2–3 times per day at polar regions. The spatial resolution has been further increased to 3.5x5.6 km² (near nadir) starting from 6 August 2019 (Rozemeijer and Kleipool, 2019).

By means of Differential Optical Absorption Spectroscopy (DOAS) (Platt and Stutz, 2008) OClO slant column densities (SCDs) have been retrieved from TROPOMI measurements (Puķīte et al., 2021). The global spatial coverage of TROPOMI,
its high spatial resolution and sensitivity with a low detection limit for OClO SCDs even at high solar zenith angles (SZAs) enable to assess the evolution of chlorine activation in unprecedented detail. The detection limit and thus the SZA threshold, for which enhanced OClO abundances might be detected, vary from instrument to instrument. Further it varies with SZA due to different signal to noise ratio, also different statistical processing like averaging over certain space and time intervals may change it. A detection limit of about $0.5–1\times10^{14}$ cm$^{-2}$ have been estimated at SZA of 90° for SCDs gridded on a resolution
of $20\times20$ km$^2$ which is well suited for measurements in the stratosphere. We can retrieve OClO slant column densities (SCDs) with a typical detection limit below $2\times10^{13}$ cm$^{-2}$ for the $20\times20$ km$^2$ area down to 65° SZA. Furthermore, the occurrence of OClO in the stratosphere ensures that no cloud filtering needs to be applied because no shielding by tropospheric clouds is expected.

The aim of this paper is to compare the spatio-temporal evolution of the retrieved OClO SCD dataset with meteorological
conditions and PSC observations in both hemispheres. European Centre for Medium-Range Weather Forecast (ECMWF) ERA5 data (Hersbach et al., 2018) are used in the comparison. We relate the OClO SCDs to the key meteorological parameters driving the chlorine activation: first, temperature, in particular with respect to the expectation that OClO appears to be produced when temperatures drop below $T_{NAT}$ along with the expected occurence of PSCs; second, potential vorticity (PV), with the expectation that OClO is being produced within the polar vortex. PV is conserved for a given air parcel in an adiabatic system
or, in other words, air parcels with different PV values do not mix adiabatically. Absolute values of PV increase in direction and towards the centre of polar vortex allowing to distinguish between air masses outside and inside the vortex. We compare OClO SCDs also with CALIPSO Cloud-Aerosol Lidar with Orthogonal Polarization (CALIOP) polar stratospheric cloud (PSC) observations. In these comparisons in the first place the initial period of the potential chlorine activation is of large interest, since we can see even localized activation events. Also the deactivation period is of great interest.

The article is structured as follows: in Sect. 2 the methodology for comparing the meteorological parameters and the TROPOMI OClO SCDs are introduced. In Sect. 3 the methodology for comparison of the TROPOMI OClO SCDs with CALIPSO PSCs dataset are described. Section 4 analyses the time series introduced in the previous sections. Finally, Sect. 5 draws some conclusions.

## 2   Relating meteorological parameters with OClO SCDs

The ECMWF data are output to the temporal resolution of 6h and are interpolated to the resolution of $1°\times1°$ in latitude and longitude during the dissemination process before further processing to ensure that our local data storage possibilities are not overburdened. It should be noted that a limited resolution can lead to uncertainties with respect to the true small scale

temperature variations. For some special mountain wave events, which can lead to mountain wave PSCs formation (Voigt et al., 2003), consequently playing a role for chlorine activation, deviations between ECMWF and models that are built to resolve the topography which induces mountain waves of up to around 10 K have been reported (e.g. Kühl et al., 2004a; Maturilli and Dörnbrack, 2006; Kivi et al., 2020).

OClO SCDs for SZAs between 89° and 90° during different winters are analysed. This SZA range is motivated by a larger ratio between the OClO SCDs and the detection limit in this range, i.e. for smaller SZA the amplitude of the observed OClO SCDs decreases faster with decreasing SZA than the detection limit does. Similar ranges (around SZA of 90°) are used in previous studies e.g. by Kühl et al. (2004b) and Hommel et al. (2014). Although given the better performance of TROPOMI, it would be possible to investigate also lower SZAs. Such an investigation, however, is beyond the scope of this study.

Time series of OClO SCD daily averages and maximum values for SZA between 89 and 90° during different winters are obtained. The maximum OClO SCD $S_{max}$ is defined as follows:

$$S \sim \mathcal{N}(\mu, \sigma^2) \tag{1}$$

$$S_{max} = P_{99}(S) - P_{99}(\mathcal{N}(0, \sigma^2)) \tag{2}$$

The 99th percentile $P_{99}(S)$ for OClO SCDs $S$ of a given day is calculated. Also the standard deviation $\sigma$ for the OClO SCDs is obtained. The 99th percentile is obtained also for the Gaussian distribution $\mathcal{N}(0, \sigma^2)$ which is parameterized by zero mean and the standard deviation $\sigma$ as obtained for the OClO SCDs. Finally the 99th percentile of the Gaussian distribution is subtracted from the 99th percentile of the OClO SCDs. It is assumed that in this way most of the surplus of the random component to the maximum is removed.

The OClO SCDs are compared with meteorological information, namely, the minimum polar hemispheric temperature $T_{min}$ (mimumum temperature for latitudes above 60°), the area where temperature is below $T_{NAT}$ and the polar vortex area. The time series of $T_{min}$ and the area where temperature is below $T_{NAT}$ are resolved in potential temperature (PT) for the lower middle startosphere. The time series of the polar vortex area are calculated at 475 K PT level.

Additionally to enable a more detailed analysis, the assigment of the meteorological quantities to the OClO SCDs for 89°<SZA<90° is obtained by a trilinear interpolation in latitude, longitude and time to the TROPOMI line of sight coordinate at 19.5 km of altitude. No radiative transfer modelling is applied during the assignment. Radiative transfer effects indicate that the mass centre of the sensitivity area of the measured OClO SCDs is expected to be located towards the direction of the Sun from the line of sight coordinate . The consideration of the radiative transfer would require a-priori constraints about the spatial variability of the OClO number density. Given its high variability and also the dependence of radiative transfer modelling on additional constraints on the atmospheric state, especially also the highly variable PSC distribution, it would introduce additional uncertainties. We have found in sensitivity studies (see Appendix A) that this displacement is expected to be less than 100 km and typical PSC concentrations do not largely affect it. It is thus below the resolution of the applied meteorological data set and the systematic effect on the performed comparison is estimated as rather limited (variation in temperature of 1 K and below and in potential vorticity of 5 PVU or below), therefore not affecting the findings of the study.

The meteorological quantities (temperature and potential vorticity) are considered here at 475 K PT level which roughly corresponds to an altitude of 19–20 km and to which we assume the retrieved OClO SCDs are most sensitive to. Selecting this level we follow earlier studies (Wagner et al., 2001, 2002; Kühl et al., 2004b) where a strong anti-correlation between minimum temperatures and OClO SCDs has been found for this PT level. The altitude corresponds well to the peak of the ozone number density profile at high latitudes (Yang and Liu, 2019). At the chosen SZA range (89-90°) the measurements also show a very high sensitivity to the investigated altitudes.

The obtained correlative dataset is then analysed resolving it with respect to the different parameters (longitude, temperature and potential vorticity).

For the daily mean OClO SCDs the random error typically is negligible, thus the systematic error component (being up to around $2\times10^{13}$ cm$^{-2}$ as estimated in Pukīte et al. (2021)) can be taken as a detection limit. For the plots resolving the OClO SCDs in longitude the standard deviation of the gridded mean is typically $\sim1\times10^{13}$ cm$^{-2}$ and occasionally $\sim2\times10^{13}$ cm$^{-2}$. The OClO SCDs gridded with respect to temperature have random uncertainties below $1\times10^{13}$ cm$^{-2}$ varying in a broad region around $0.5\times10^{13}$ cm$^{-2}$, with larger values for days with larger temperature variability within the 89°<SZA<90° band. The OClO SCDs resolved with respect to the potential vorticity have even lower random uncertainties ($\sim0.2\times10^{13}$ cm$^{-2}$), only at the minimum and maximum PV values the standard deviation can reach $\sim1$-$1.5\times10^{13}$ cm$^{-2}$.

Given that also here the systematic error component is mainly dominating, the detection limit thus is expected to be below $\sim2.5\times10^{13}$ cm$^{-2}$ with systematic error as the dominating source of the uncertainty.

## 3   CALIOP PSC observations

In addition, we relate the retrieved OClO SCDs with the Level 2 Polar Stratospheric Cloud provisional version 1.10 product (Pitts et al., 2009). The PSC product, freely provided by (NASA/LARC/SD/ASDC, 2016), is retrieved from the Cloud-Aerosol Lidar with Orthogonal Polarization (CALIOP) observations on Cloud-Aerosol Lidar and Infrared Pathfinder Satellite Observations (CALIPSO) satellite. From the CALIOP PSC product we use the provided PSC cloud mask profiles indicating whether a PSC is detected above a certain location as a function of altitude. The advantage of the use of the PSC mask product in our opinion is that it reduces possibility to misinterpret the aerosol information which would be the case if backscatter data would be used instead. We neglect the available distinction with respect to different PSC types as the aim of the current study is to check how the general existence of PSCs relates with the OClO SCDs we have measured. We also consider the detection sensitivity which is provided in the PSC product where the horizontal averaging which was necessary to detect PSC is provided. To be able to match an OClO SCD at a given location which is not altitude resolved with a single piece of information about PSCs, we merge the PSC existence profile information as well as the altitude resolved detection sensitivity to a single generic

quantity. This quantity, which we call PSC evidence $E$ in the following and which up to our knowledge have not been used in the literature so far, is calculated as a sum of the PSC signals originating from all different altitudes at a given location:

$$E = \sum_i \frac{M_i}{A_i} \tag{3}$$

where $M_i$ is a boolean being unity if a PSC is reported in the CALIOP data at an altitude level $i$ more than 4 km above the tropopause. $A_i$ is the reported horizontal averaging being either 1, 3, 9 or 27 corresponding to the horizontal averaging of 5, 15, 45 or 135 km, respectively, which was necessary to detect the PSC.

For the comparison, each CALIOP measurement is collocated with the average of TROPOMI measurements within the range of 89°<SZA<90° on the same day that are less than 100 km away. It is done because of the larger spatial coverage of TROPOMI as well as to largely eliminate random error contribution of individual TROPOMI measurements.

In addition, also daily mean and maximum evidences are obtained from PSC evidences calculated beforehand for all CALIOP measurement locations above 60° latitude. While the collocated PSC evidences describe the PSC existence at and near the analysed TROPOMI measurements, these two additional parameters provide additional information about PSC extent in the whole polar region.

Moreover, we performed a sensitivity study which revealed that the PSC evidence is better suited as an indicator of the presence of PSCs than the mean backscatter ratios, especially for low level PSCs. Details about the sensitivity study are given in Appendix B.

## 4 Interpretation of the TROPOMI OClO measurements with respect to meteorological quantities and CALIOP PSC observations

### 4.1 Arctic winters

#### 4.1.1 Winter 2017/2018

The first winter (2017/2018) after TROPOMI was launched was a rather cold stratospheric winter especially with cool temperature anomalies in January until the beginning of February over the polar cap (Wang et al., 2019). A sudden stratospheric warming event has been reported for 12 February characterized by a polar vortex split (Butler et al., 2020; Hall et al., 2021).

For this winter unfortunately many days of measurements are missing due to calibration processes. The time series of OClO SCDs daily averages for SZA between 89 and 90° during this winter are plotted in the top panel of Fig. 1. The averages are shown for all data (blue), data within the polar vortex with PV>35 PVU at the PT level of 475 K (green), also the maximum OClO SCD $S_{max}$ is plotted (red). In the second panel, the latitudes of the TROPOMI pixels which contributed to the OClO SCDs are illustrated (left axis). In this panel also the size of the polar vortex area is plotted being defined as the area with PV>35 PVU at PT 475 K. The two lower panels provide relevant meteorological information: time series of the (northern)

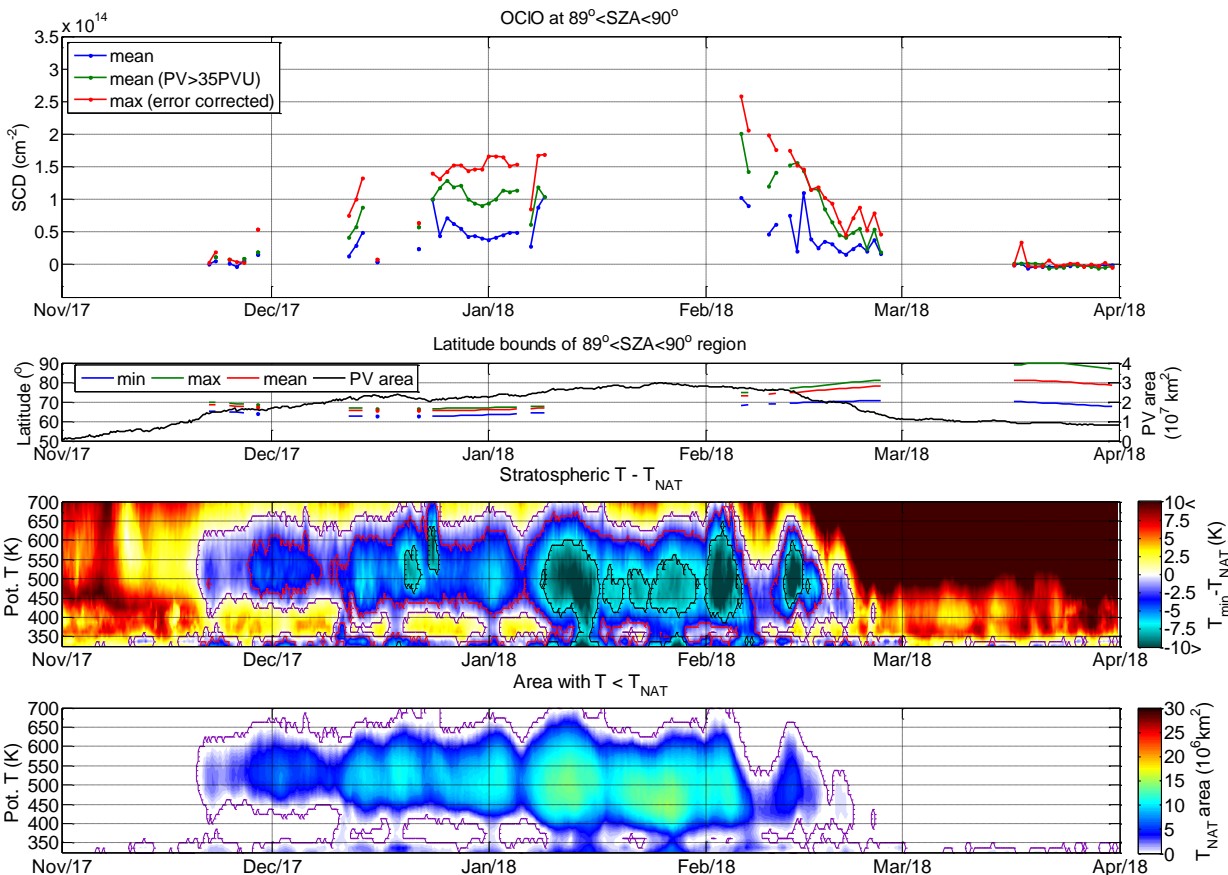

**Figure 1.** Time series of daily OClO SCDs for the Arctic winter 2017/2018 in comparison with the meteorological quantities. Please note that many days of measurements are missing for this winter due to calibration processes after launch. Top panel: The blue line represents the mean daily OClO SCDs for 89°<SZA<90°, the green line the mean of the measurements within the polar vortex (PV >35 PVU at PT 475 K), and the red line the maximum OClO SCDs (for details see text). Second row: time series of minimum, maximum and mean latitudes of the TROPOMI pixels which contribute to the mean OClO SCDs shown in the top panel (left axis). Also shown is the polar vortex size (area where PV >35 PVU at the PT 475 K) indicated by a black line (right axis). Third row: Time series of temperature evolution in the lower stratosphere represented as difference between the minimum and NAT condensation temperature ($T_{NAT}$) as function of altitude (indicated by the potential temperature). Violet, red and black contourlines lines indicate $T_{NAT}$, $T'_{NAT}$ and the ice freezing temperature $T_{ICE}$, respectively. Bottom row: Time series of size of the area where the temperature is below $T_{NAT}$ as function of the potential temperature. Zero is indicated by the violet contourline.

hemispheric minimum temperature expressed as the difference between temperature and $T_{NAT}$ as function of the PT. In the
bottom panel the area where temperature is below $T_{NAT}$ is plotted, with the violet line showing the boundary of this area.

Additionally in Fig. 2 the temporal variation of the OClO SCDs for 89°<SZA<90° is presented resolved with respect to different parameters (longitude, temperature and PV) to allow for a more detailed analysis. The top panel resolves the SCDs in

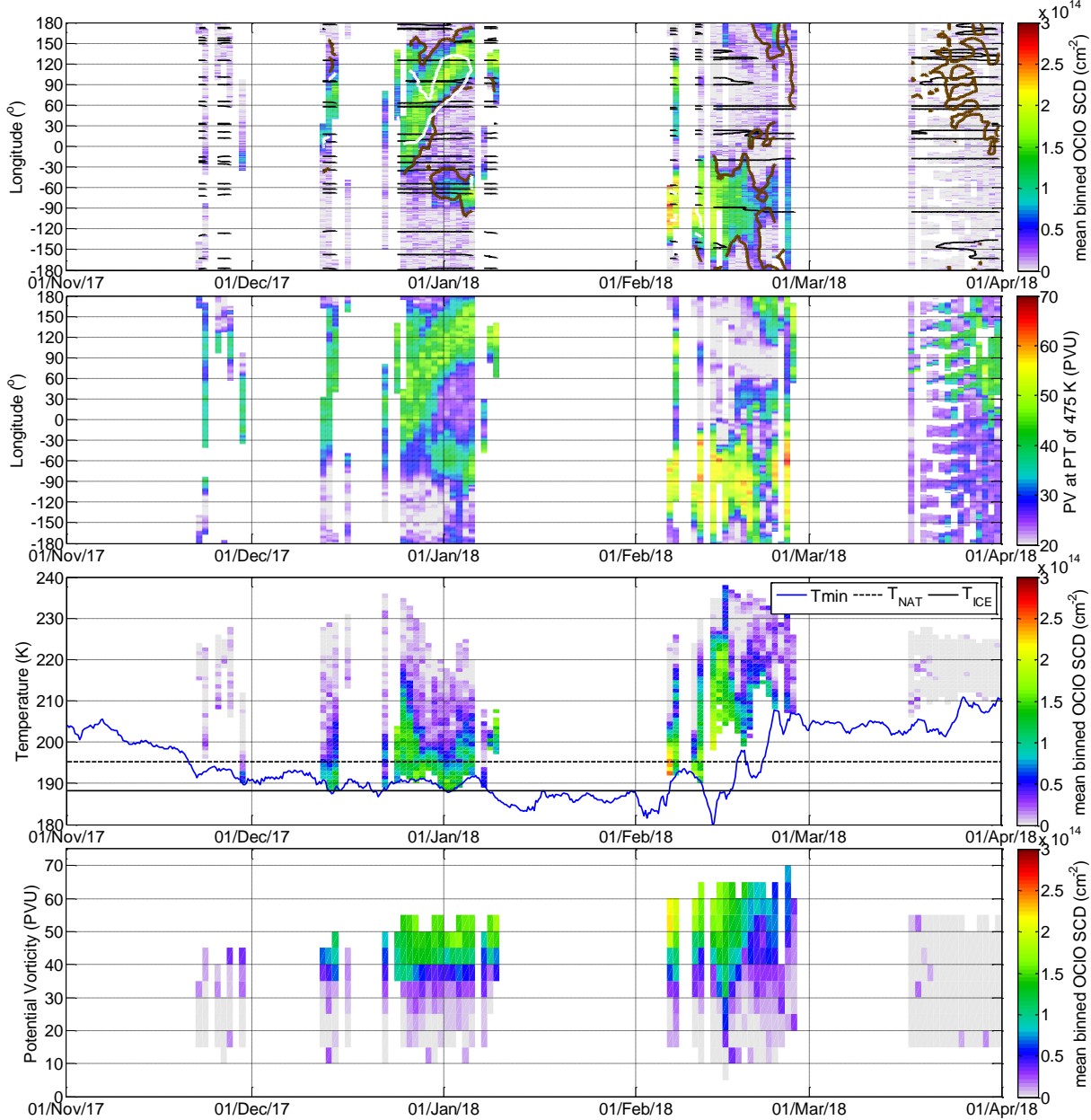

**Figure 2.** Top panel: Time series of the daily measured OClO SCDs for 89°<SZA<90° resolved longitudinally (resolution 1°, positive values – East longitudes, negative values – West longitudes) for the Arctic winter 2017/2018. Black, brown and white contourlines indicate the maximum surface elevation of 1 km, PV 35 PVU at PT 475 K and temperature $T_{NAT}$, respectively. Second panel: Time series of the potential vorticity at the location of the OClO measurements shown in the panel above. Third panel: The same OClO dataset as in the top panel but resolved as function of temperature (resolution 1 K) at the PT 475K level. Here also the minimum polar hemispheric temperature (mimumum temperature for latitudes above 60°) at this potential temperature level (blue line) and the values of $T_{NAT}$ and $T_{ICE}$ (at 19.5 km) are indicated. Bottom panel: Same OClO dataset as in the top panel, but resolved as function of the potential vorticity (resolution 5 PVU) .

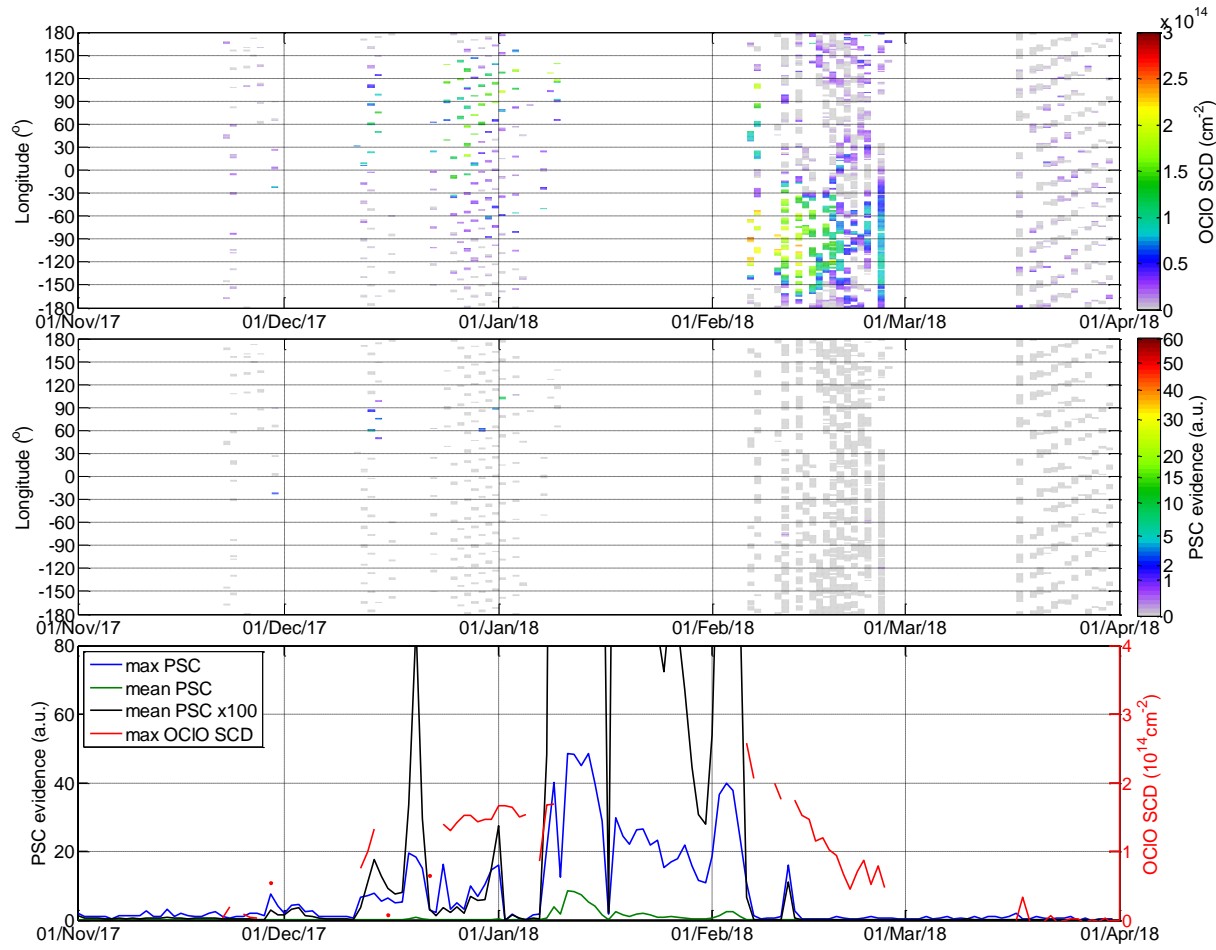

**Figure 3.** Top panel: Time series of OClO SCDs for 89°<SZA<90° being collocated to CALIOP measurements and longitudinally resolved (resolution 1°, positive values – East longitudes, negative values – West longitudes) for the Arctic winter 2017/2018. Middle panel: Time series of the CALIOP PSC evidence collocated to the OClO SCDs in the top panel. Bottom panel, left axis: time series of maximum and mean PSC evidence for latitudes above 60° (blue and green lines, respectively), mean PSC evidence derived from the CALIOP PSC mask product scaled by 100 (black line); right axis: maximum OClO SCDs (red line).

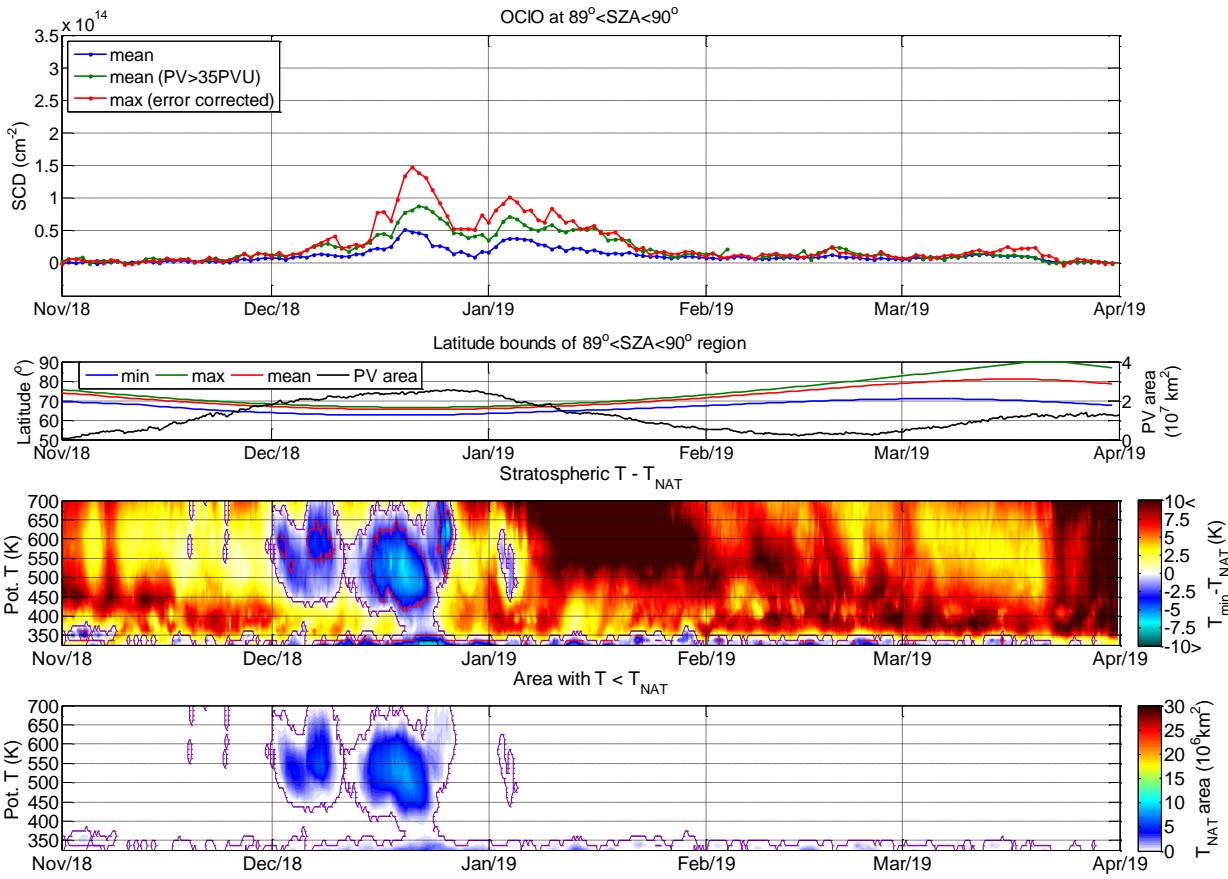

**Figure 4.** Same as Fig. 1 but for the Arctic winter 2018/2019.

longitude (1° grid). The contours are plotted for areas with local temperature below $T_{NAT}$ (white), the polar vortex boundaries
(PV>35 PVU at the potential temperature level 475 K, brown) and for a maximum surface elevation of more than 1 km above
220 the sea level (black). The second panel from top provides the complete PV information at the potential temperature 475 K at the
place of the measurements of the top panel. The third and fourth panels from top resolve the data with respect to temperature
at the measurement location (on 1 K grid at the PT level of 475 K), as well as with respect to the PV (on 5 PVU grid) at the
same level. In the third panel from top, lines indicating $T_{NAT}$, $T_{ICE}$ and minimum temperature (at 19.5 km altitude) are added.

Time series of the PSC evidences resolved in longitude (on a 1° grid) are shown in the middle panel of Fig. 3. The plots
for the respective collocated OClO SCDs are shown in the top panel. The gridded data are shown only for grid points where
at least 100 TROPOMI measurements have contributed in order to ensure low random error contribution. Mean and maximum
PSC evidences calculated for all CALIOP measurements at latitudes for polar areas of the respective hemispheres above 60°
are plotted in the bottom panel, (x-axis) along with the daily maximum OClO SCDs (y-axis).

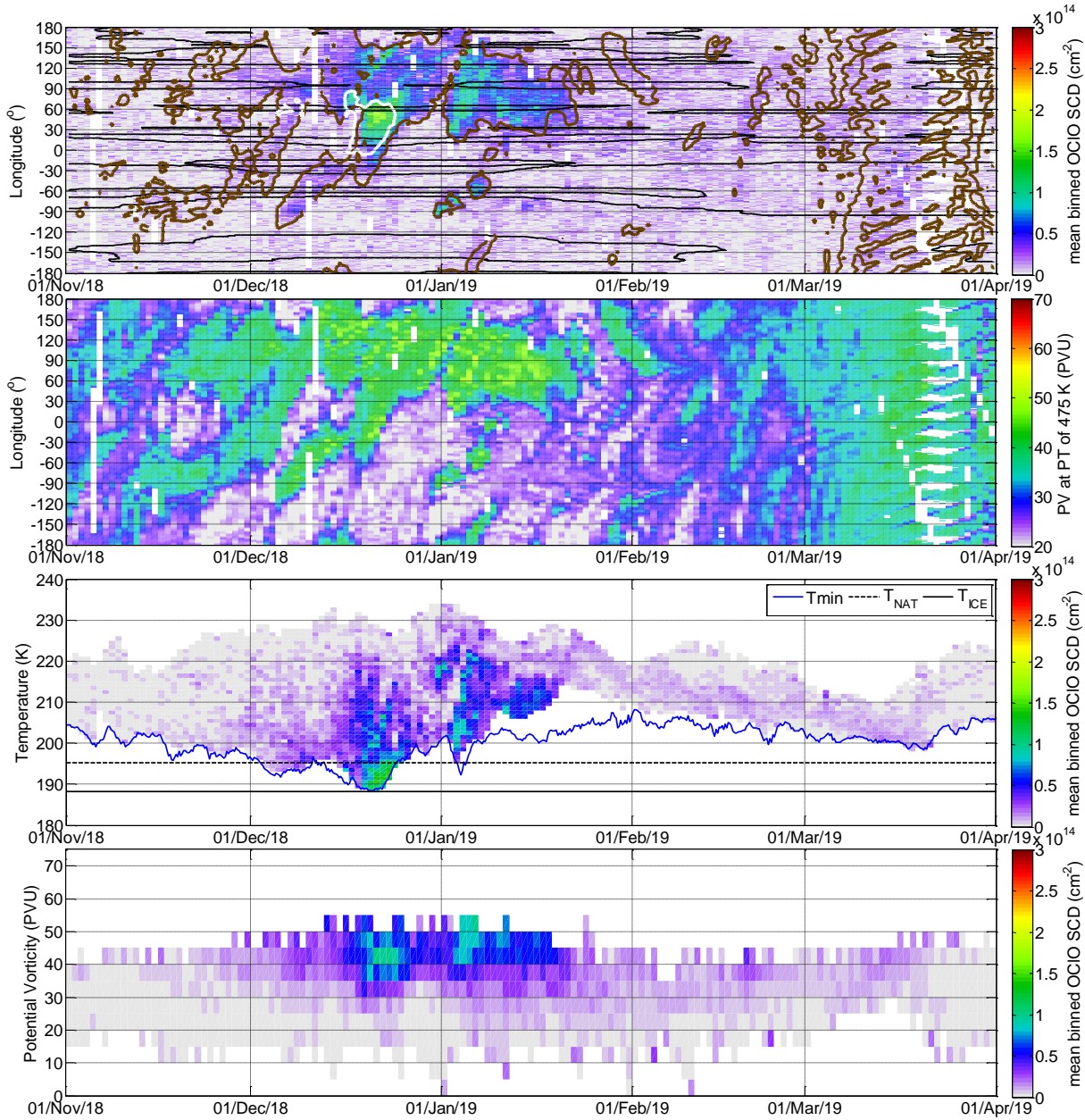

**Figure 5.** Same as Fig. 2 but for the Arctic winter 2018/2019.

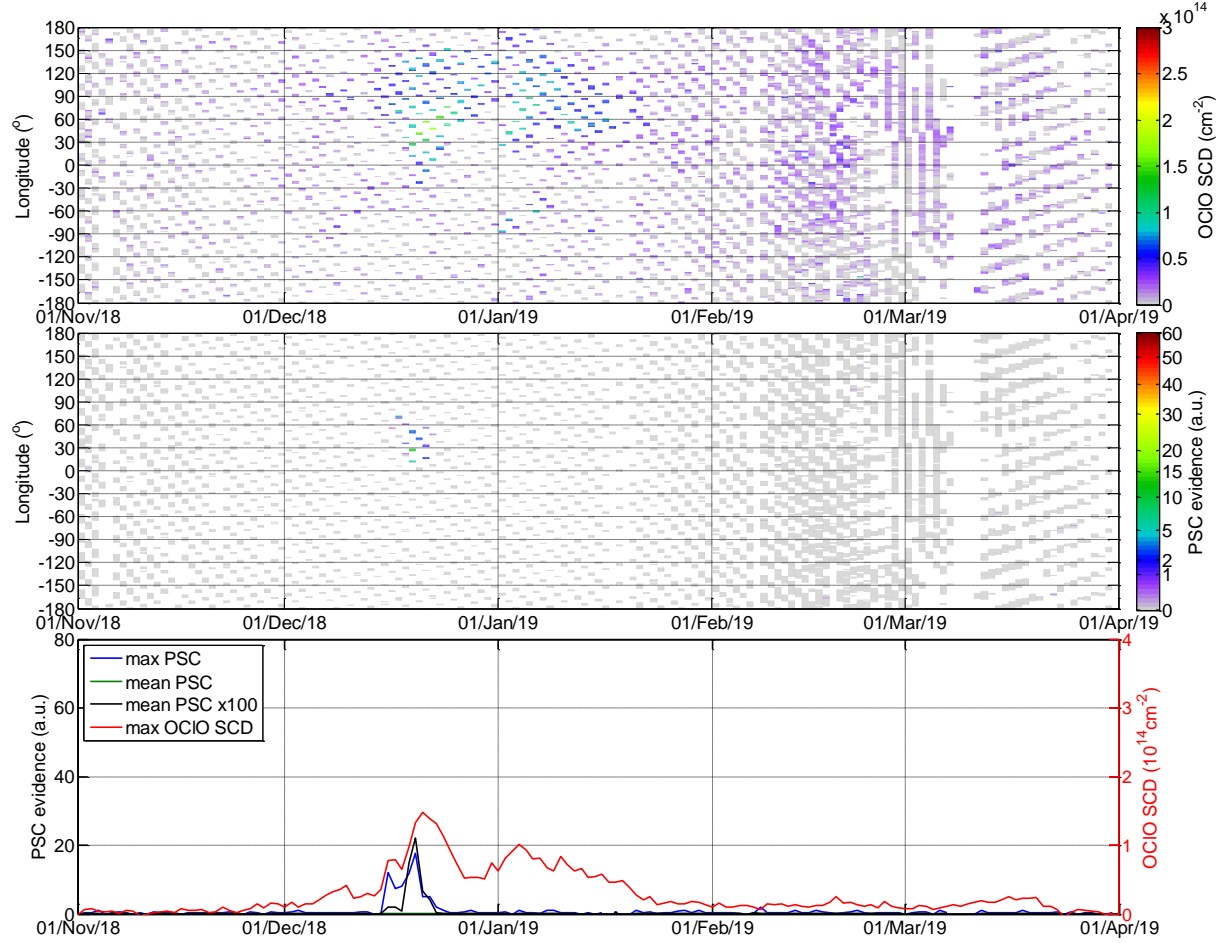

**Figure 6.** Same as Fig. 3 but for the Arctic winter 2018/2019.

23 November 2017 is the first day we were able to retrieve OClO SCDs with almost complete longitudinal coverage. Al-
though the minimum hemispheric temperature is slightly below $T_{NAT}$ we do not see an increase in the OClO SCDs. However, a
clear increase is observed on 29 November 2017 above the same area. In this case a temperature below $T'_{NAT}$ is observed locally
at the measurement area as it can be deduced from Fig. 2, third panel, showing increased OClO SCD values at local tempera-
tures around and below $T'_{NAT}$. Thus a chlorine activation process at the locations of the measurements at the 89°<SZA<90°
can be expected. There is still a possibility that already somewhere else activated air masses have been transported into the
analysed measurement region, however the CALIOP data (Fig. 3) also show an evidence of PSC formation at longitudes around
20° W which perfectly matches with the location of the increased OClO SCDs on that day, providing a strong evidence of the
chlorine activation at this location. For the next available days (12–14 December 2017) even more enhanced OClO SCDs are
measured. They are observed almost only within that part of the polar vortex where the temperatures are below $T_{NAT}$. The
region extends for longitudes between 0° and 120° E. The region where PSCs are evident is slightly smaller (40° and 110°

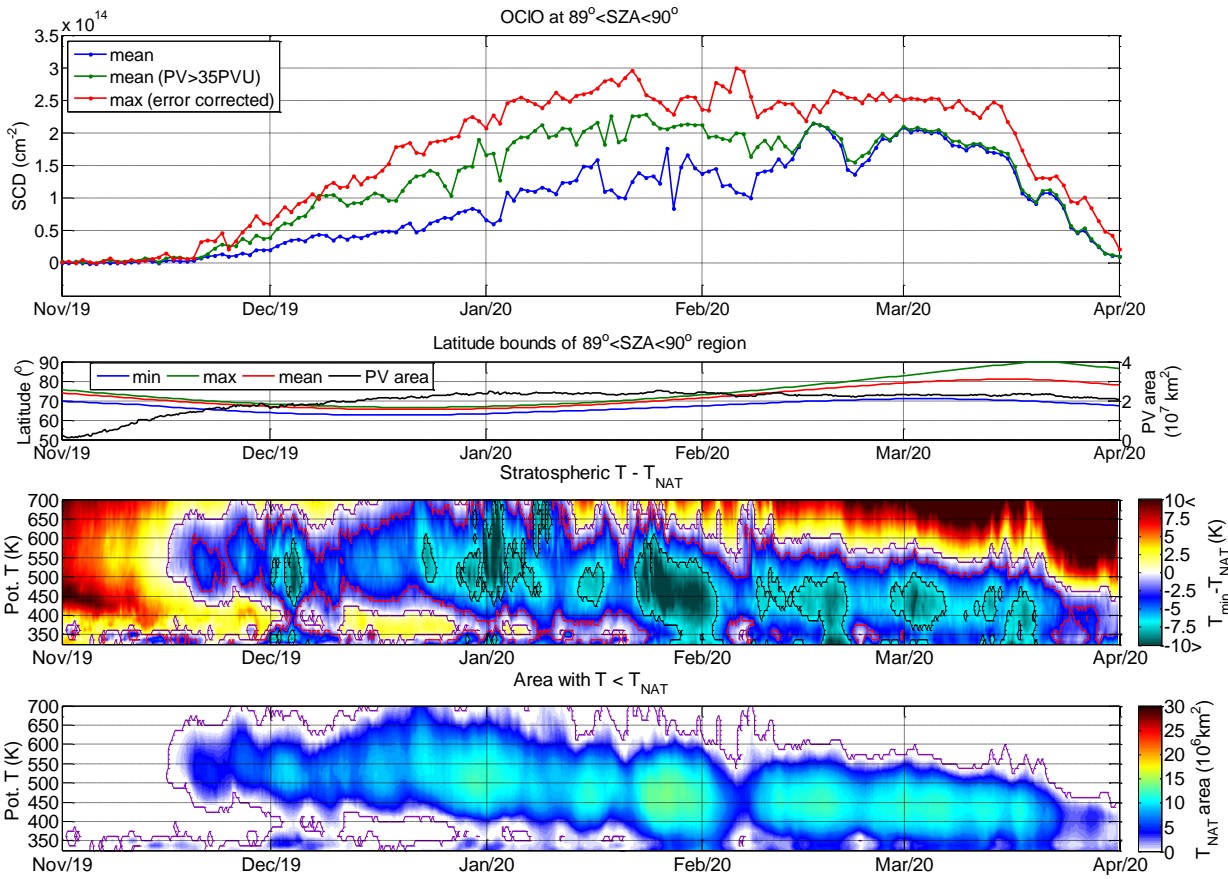

**Figure 7.** Same as Fig. 1 but for the Arctic winter 2019/2020.

E) suggesting that the enhanced OClO SCDs observed outside this region are either caused by chlorine activation on previous days or due to mixing. For the more eastern regions, still within the vortex, no OClO can be seen, indicating that the observed OClO is still rather fresh and is not yet well mixed with the air masses of the whole vortex. This is not anymore the case around the next available period after Christmas 2017 where enhanced OClO SCDs are observed within the whole vortex and also at temperatures well above $T_{NAT}$ which corresponds to a period of a slight vortex warming. PSCs are evident in this period

only for few instances tending to confirm that the bulk of chlorine activation happened earlier. A persistent polar vortex exists until the first week of February with OClO well distributed within the polar vortex as visible for the days when measurements are available. Also the minimum temperature is below $T_{NAT}$ for almost all of this time. The seasonal maximum SCD in the presented data is observed at the beginning of February 2018. However, PSC evidence is zero for the collocated CALIPSO measurements. Mean and maximum PSC evidences within the polar region are largely reduced which is plausible (because

temperature has risen above $T_{ICE}$ dissolving ice aerosol) with respect to the previous days for which no OClO measurements were available. Nevertheless the local temperature is around $T'_{NAT}$ (i.e. well below $T_{NAT}$), thus we do not have an explanation

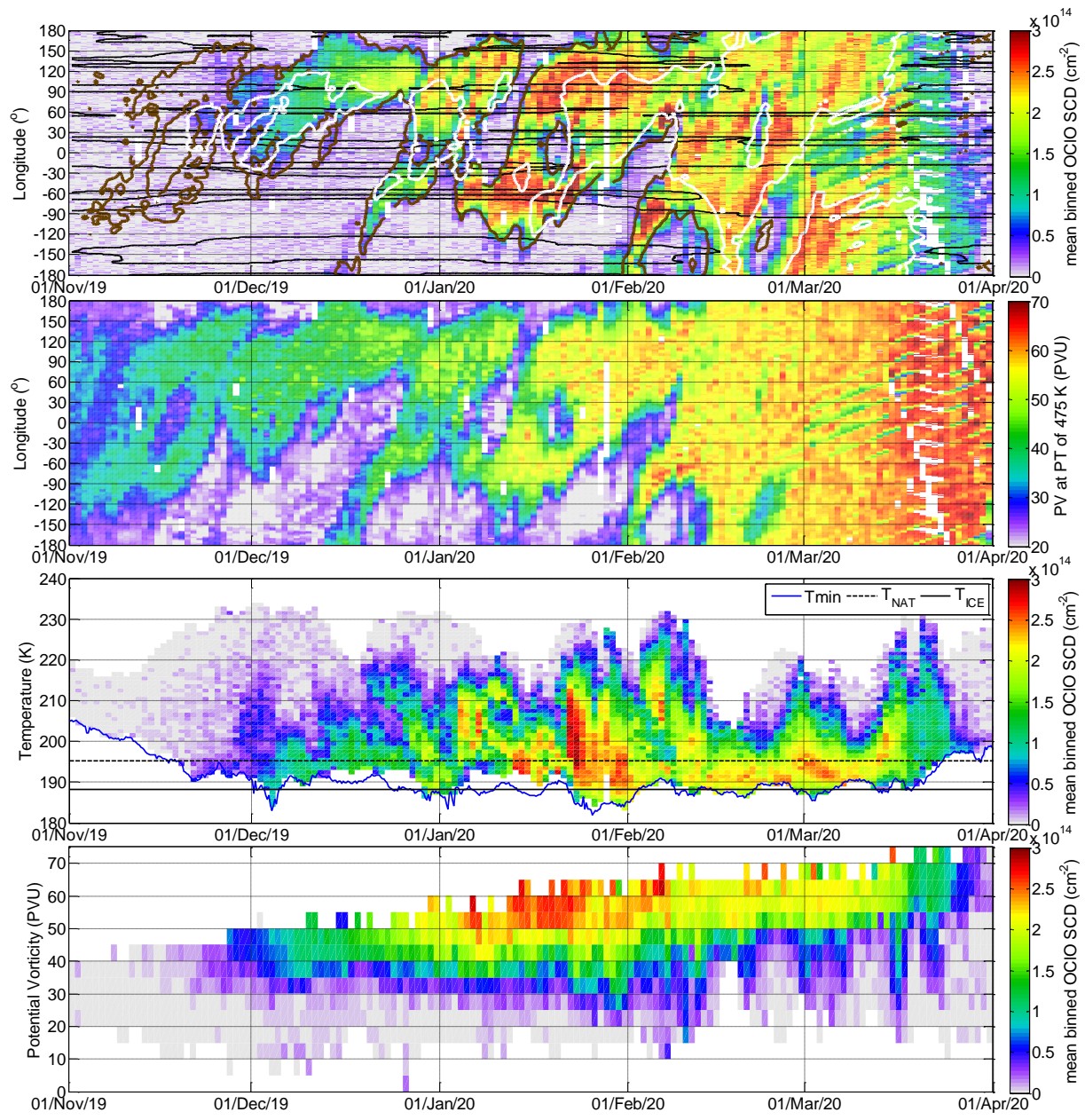

**Figure 8.** Same as Fig. 2 but for the Arctic winter 2019/2020.

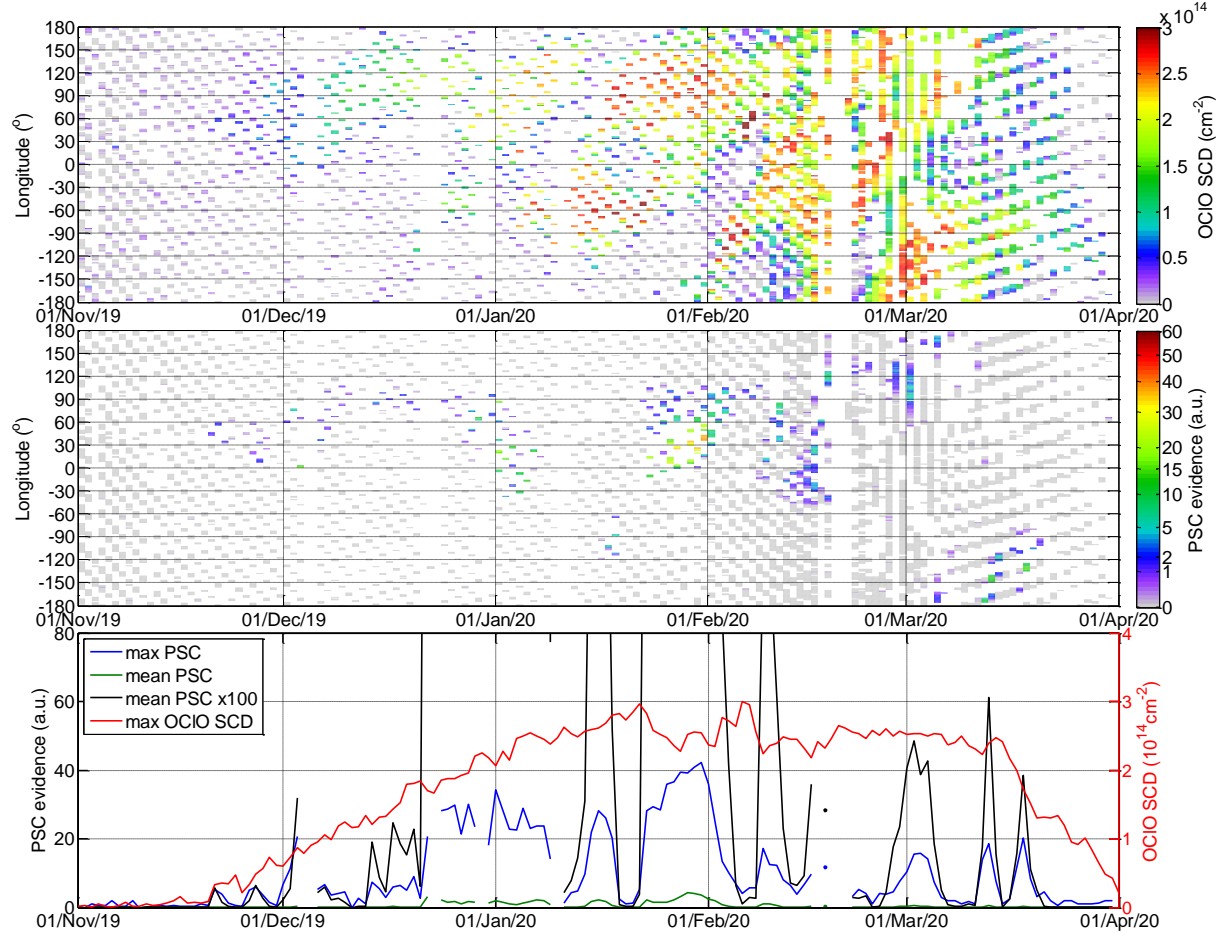

**Figure 9.** Same as Fig. 3 but for the Arctic winter 2019/2020.

of the complete lack of NAT and STS PSCs for this time period. A sudden stratospheric warming took place on 12 February with a vortex split (Butler et al., 2020; Hall et al., 2021). At the end of the second week the minimum temperature drops again below $T_{ICE}$ before which the vortex area seems to have stayed rather constant for a few days (Fig. 1, second plot from top).

Nevertheless the OClO values continue to decrease afterwards, the temperature gradient becomes quite large within the split vortex which can be deduced by the increased OClO at high temperatures in the temperature resolved time series of OClO SCDs (third panel in Fig. 2). After this short cooling the temperature rises rapidly, the vortex area decreases and the OClO SCDs continue to decay. The breakup of the polar vortex is also evident in the bottom plot of Fig. 2 where still increased OClO SCDs are found towards lower PV values. A second similar event, but not as strong, is observed at the last days in February

(26 February). Here also PSCs are barely evident at the longitudes (around 120° W) at which largest OClO SCDs are observed. The vortex eventually strengthens again at the beginning of March when mean zonal winds become westerly again (Butler et al., 2020) but it has no relevance for chlorine activation because of the high temperatures.

### 4.1.2 Winter 2018/2019

The following winter 2018/2019 has been reported as being unusual in terms of the polar vortex variability (Lee and Butler, 2020): with both a major sudden stratospheric warming and a reformation of a strong vortex later. In terms of minimum temperature (see third plots from top in Figs. 4 and 5, for technical explanation of plots please see the description for the previous winter) the beginning of the winter was rather warm, the temperatures dropped below $T_{NAT}$ only in December. However the mean OClO SCDs (Fig. 4, upper plot) appear to be slightly but consistently increased above zero already during the last days of November with enhanced OClO SCDs above Greenland and Northern Asia (upper plot in Fig. 5). This increase however technically is still below the detection limit of $2\times10^{14}$ cm$^{-2}$. An OClO production in the area covered by the plotted SZA range (89°<SZA<90°) can likely be excluded because no OClO enhancements at the lowermost temperature bins in the temperature resolved time series of OClO SCDs are found (Fig. 5, third panel). This finding does not exclude that such an activation could have taken place in some other area not covered by the SZA range investigated here. Lee and Butler (2020) report a begin of the increase of a vertically propagating wave activity during November and thus local drops of the temperature below $T_{NAT}$ induced by mountain waves could have been a possibility for OClO formation because the minimum temperature at 600 K reaches $T_{NAT}$ in that period. The CALIOP data (Fig. 6) however do not show any evidence of PSC formation.

The mean OClO SCDs increase further at the beginning of December a few days after the temperature dropped below $T_{NAT}$. This delay probably indicates that the area where this drop occurs is small or that the drop was not sufficient to overcome the supersaturation limit for the PSC build up. The OClO SCDs are increased for both the areas within the polar vortex as well as for areas of lower temperature (Fig. 5). The OClO SCDs show a clearer increase on 6 December 2018 which coincides with $T_{min}$ dropping below T'$_{NAT}$. After a small warming, the stratospheric temperatures drop once more (on 15 December) below T'$_{NAT}$ which coincides with a new strong increase in the OClO SCDs on the following day. On 16 December also the mean and maximum evidence of PSCs (Fig. 6, bottom panel) has a clear increase above zero. For some of the coldest days (17 – 24 December) the area of minimum temperatures is covered by the TROPOMI measurements in the range 89°<SZA<90°. The maximum OClO SCDs of this season are observed on 21 December. Local PSC evidence (Fig. 6, middle panel) above zero is also observed but only at a few longitudes (10°-70° E) for 17 – 21 December with a maximum at 19 December. The PSC evidence clearly corresponds to increased OClO SCDs of around $1\times10^{14}$ cm$^{-2}$ or higher (compare Fig. 6, top panel and middle panel, as well as daily mean and maximum PSC evidence values with the timeline of the maximum OClO SCDs in the bottom panel). On the other hand, such or even higher OClO SCDs not necessarily correspond to an observation of the PSC evidence above zero. The largest OClO SCDs on these days are clearly limited to the area with temperatures below $T_{NAT}$ which are located eastwards of the Scandinavian mountains and around the Ural mountains: this could be an indication for mountain waves having enhanced the chlorine activation process. The OClO SCDs in the rest of the analysed polar vortex area remain lower but well above the random uncertainty level and at or above the detection limit. Further, these look like remnants of earlier chlorine activation. After this cooling the polar vortex slowly starts to shrink (Fig. 4, second plot from top), is warmed up at the end of December (Fig. 4, third plot from top) as the prelude for an early sudden stratospheric warming event reported on 2 January (Lee and Butler, 2020). The atmospheric temperatures rise above $T_{NAT}$ on 27 December and stay slightly above

$T_{NAT}$, eventually dropping once more below it on 3 and 4 January 2019. However the area with temperatures below $T_{NAT}$ is very small for these days. The appearance of one additional OClO peak at the beginning of January can be attributed to the irregular shape of the polar vortex and to the fact that the earlier activated air masses are moved inside the $89° < SZA < 90°$ range. This interpretation is supported by the temperature resolved time series of OClO SCDs (third panel of Fig. 5) where the enhanced OClO SCDs appear at quite warm temperatures. These enhanced OClO values especially at the end of December and in January even appear for high temperatures ($> 20$ K above $T_{NAT}$). On these days also an increase of the potential vorticity (above 50 PVU) is observed (bottom panel of the same figure) which indicates that here air masses are seen which were not observed before, because they were located deep in the centre of the polar vortex. Afterwards the OClO SCDs decay until mid of January to values below the detection limit. In February and March the formation of a very strong polar vortex has been reported (Lee and Butler, 2020) but the temperatures never fell again below the threshold for the chlorine activation.

### 4.1.3 Winter 2019/2020

In the winter 2019/2020 an exceptionally strong and cold stratospheric polar vortex was formed which maintained cold temperatures for PSC formation and ozone destruction until the end of March (e.g. Lawrence et al., 2020; Weber et al., 2021). Figs. 7 and 8 show the evolution of the OClO SCDs along the cold stratospheric temperatures during the stable polar vortex in winter 2019/2020. Fig. 9 illustrates the PSC evidence from CALIOP observations. The hemispheric $T_{min}$ dropped below $T_{NAT}$ as early as on 16 November 2019, but increased OClO SCDs were observed on 21 November when $T_{min}$ was already lower than T'$_{NAT}$ (Fig. 7). In the third panel of Fig. 8 it can be further seen that this increase happened exactly when the local temperature fell below T'$_{NAT}$. Also nonzero PSC evidences (at longitudes 30°-60° E and few days later 0°-60° E) coincide with some of the increased OClO SCDs (Fig. 9). In the third panel of Fig. 8 it can further be seen that the OClO SCDs show a new enhancement when the temperatures again drop below T'$_{NAT}$ at the beginning of December. Also PSCs are reported (Fig. 9, middle panel) as evident at a few longitudes (mainly 60°-90° E). With temperatures staying at these low levels or even dropping below $T_{ICE}$ the OClO SCDs almost linearly increase until the end of the second week of January 2020. More variation can be seen in the polar mean and maximum hemispheric PSC evidences which increase by an order of magnitude whenever $T_{min}$ drops below $T_{ICE}$. This increase in the PSC evidence however seems not to have a clear relation with the observed OClO SCDs. Since mid January, with temperatures still being low, the OClO SCDs remain nearly constant at about $2.5 \times 10^{14}$ cm$^{-2}$ till mid March. During that period in several occasions (10, 20 February, 16 March) air masses with slightly enhanced OClO SCDs appear to be mixed with air from outside the polar vortex (with low PV values) (Fig. 8, bottom panel). Also the opposite happens at 21–26 February when enhanced OClO SCDs appear only at very high PV values. In the last two weeks of March the stratosphere starts to heat up, there is also no evidence of PSCs in the CALIOP data reported anymore and the OClO SCDs decrease reaching almost zero at the end of the month although there is still a small area with temperatures below $T_{NAT}$ at lower altitudes.

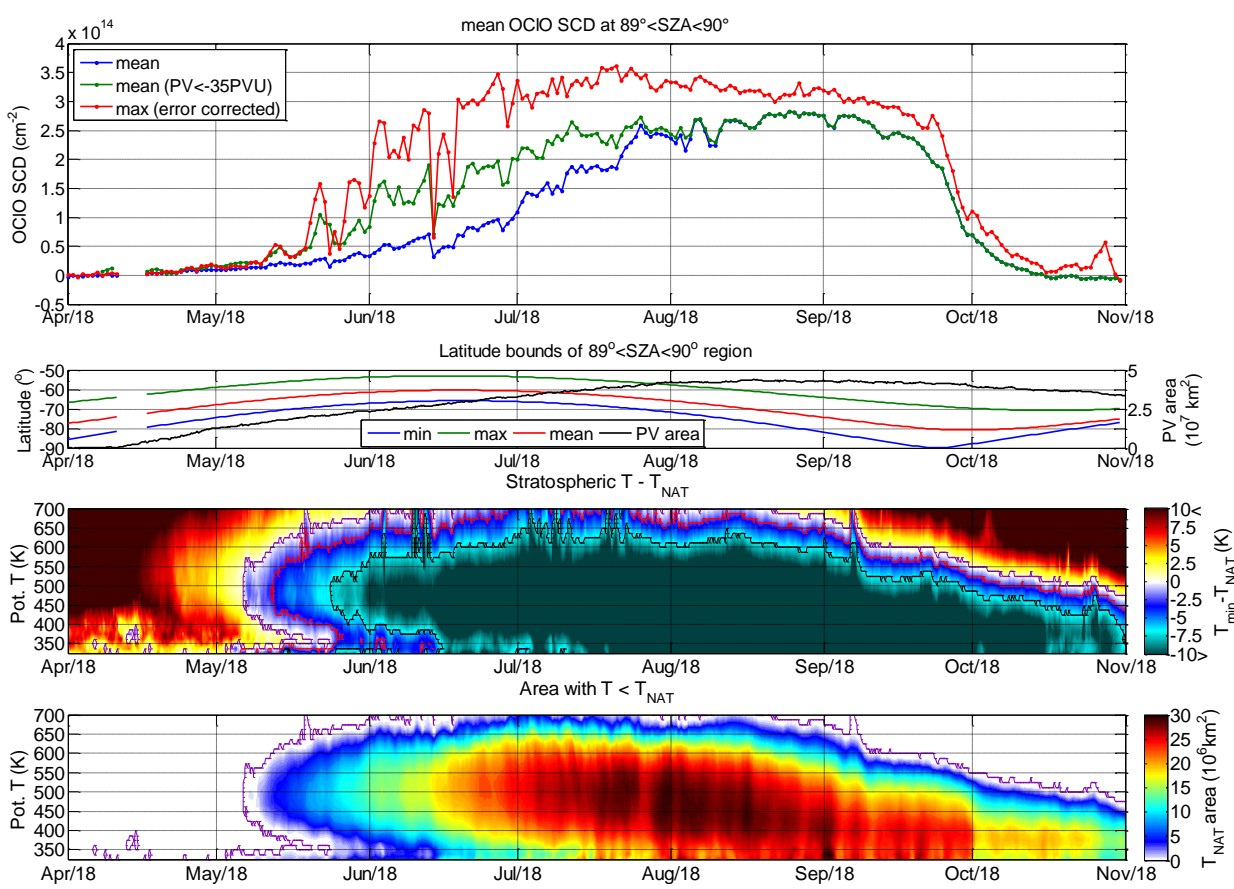

**Figure 10.** Same as Fig. 1 but for the Antarctic winter 2018.

## 4.2 Antarctic winters

### 4.2.1 Winter 2018

The Antarctic winter 2018 was relatively stable and colder in comparison to most years of the prior decade with a large and persistent ozone hole (Klekociuk et al., 2021). This accordingly resulted in an expected development of the OClO SCDs as shown in Figs. 10 and 11. For most of the season, due to the well centred shape of the polar vortex, regions with local temperatures above the hemispheric minimum temperature are observed. Only at the end of August and in September the area with 89°<SZA<90° becomes located at regions close to $T_{min}$ because then the more central parts or the vortex at higher

latitudes become illuminated.

The polar vortex starts to form in mid April (see the development of PV area in Fig. 10, second plot from top), and temperatures drop below $T_{NAT}$ in the first 10 days of May (7 May) as shown in Fig. 10, third plot from top. Shortly afterwards, the temperatures decrease below $T'_{NAT}$, and an increase in the maximum of OClO SCDs within the polar vortex is observed. This

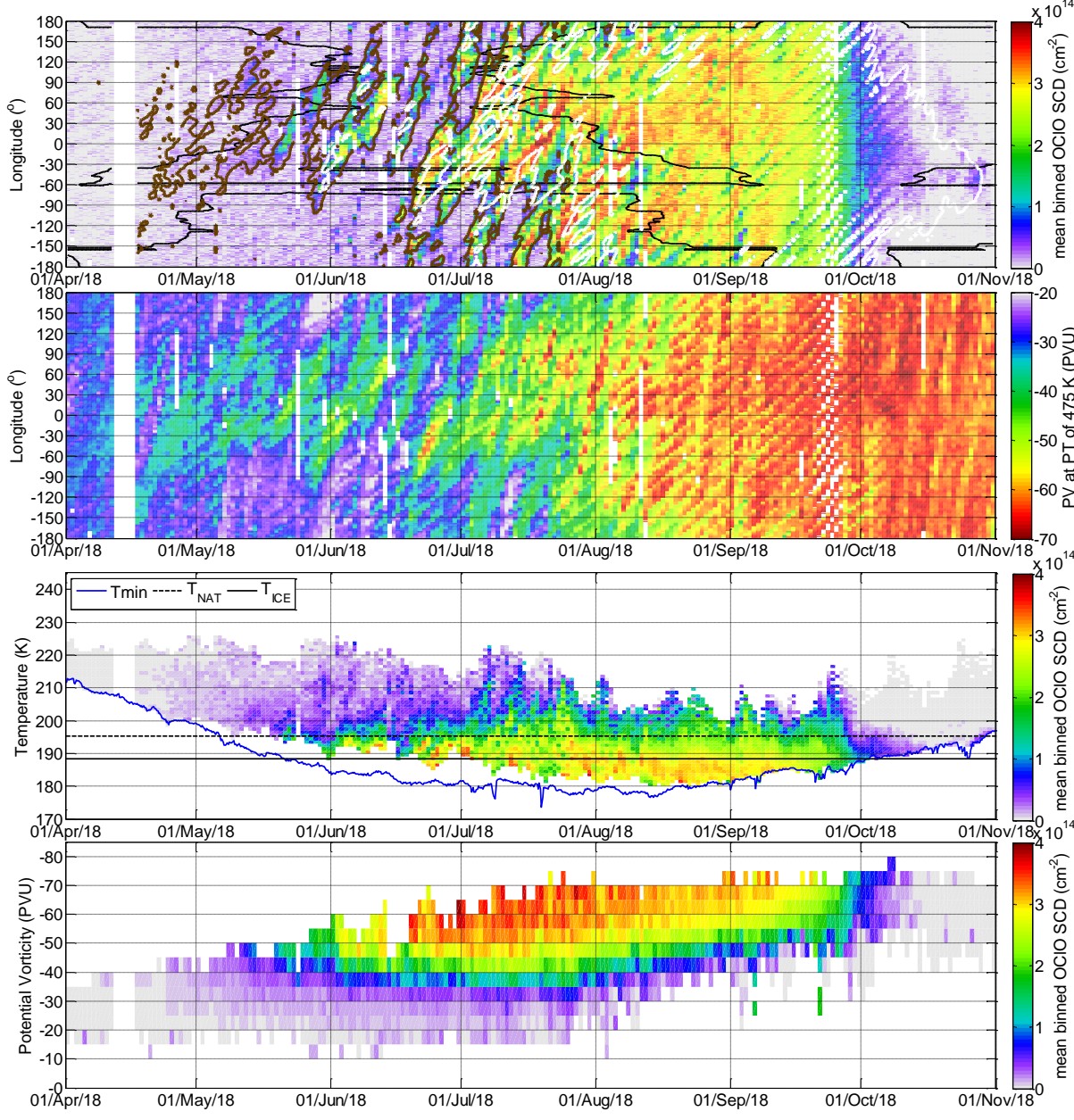

**Figure 11.** Same as Fig. 2 but for the Antarctic winter 2018 with brown line in the top panel indicating PV = -35 PVU, accordingly.

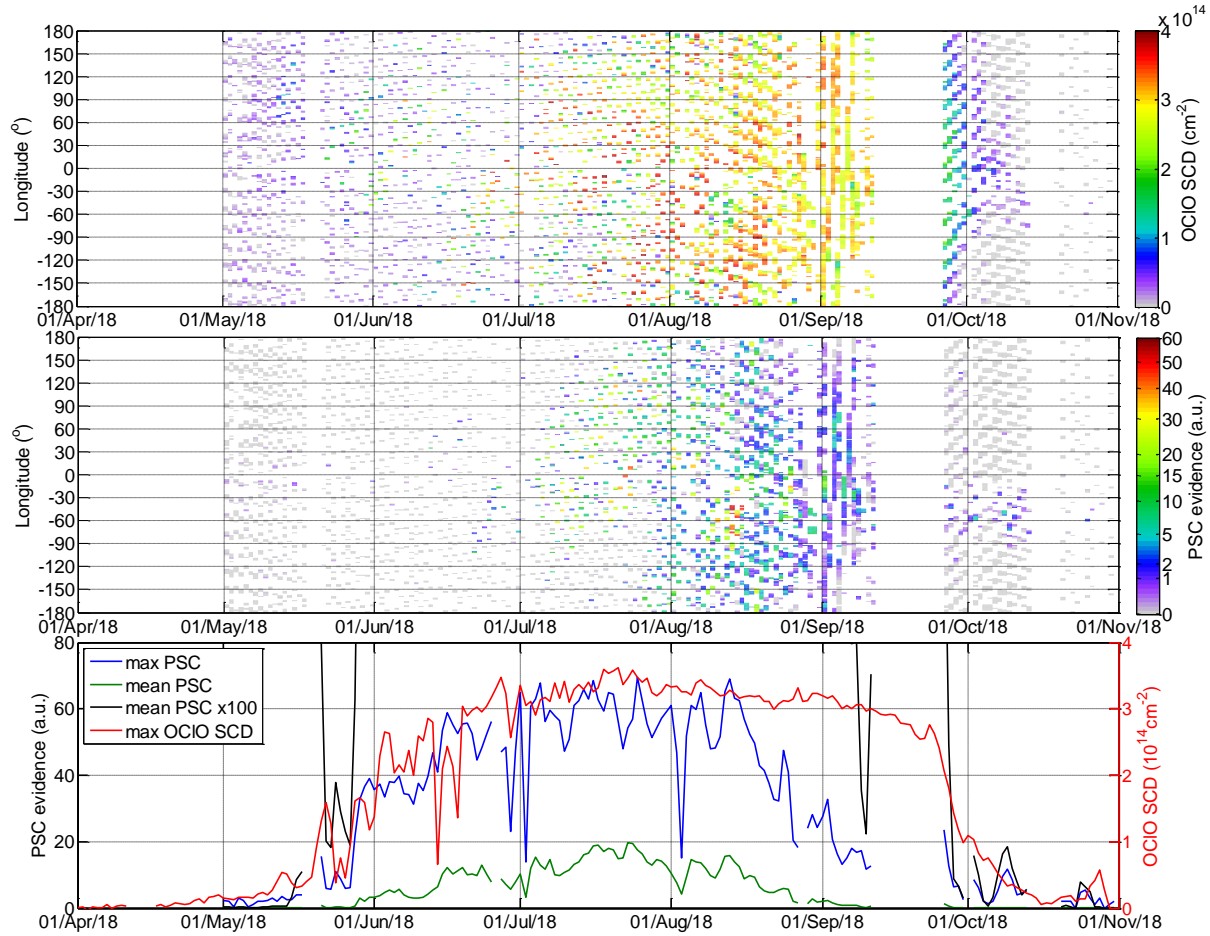

**Figure 12.** Same as Fig. 3 but for the Antarctic winter 2018.

signal can also be well identified at the largest PV values. This OClO could have been transported from regions more inside

the vortex where it is colder than in the investigated SZA region as the local temperature bins do not yet cover the temperatures

below $T_{NAT}$. An indication for a local OClO activation would however be the PSC evidence values that were slightly above

zero since the beginning of May (Fig. 12, middle panel). These values (at longitudes around 15°E - 60°W) seem however not

to have a clear relation with the collocated OClO SCDs (Fig. 12, top panel) which are larger at other longitudes (60° - 120° E)

than at the collocated longitudes. However, when also the local temperatures drop below $T_{NAT}$ (starting with 20 May), clearly

enhanced OClO SCDs appear, despite the local PSC evidence being above zero only once in these days at the end of May

and at a single longitude (10°E) where at the same time the polar mean and maximum PSC evidence increases distinctively.

Here also the time series of OClO SCDs resolved with respect to temperature shows larger OClO SCDs at temperatures close

to $T_{NAT}$. Even 'trails' with increased OClO SCDs starting at locations with elevated surface heights (black contourlines in the

longitudinally resolved time series of OClO SCDs plot in Fig. 11) and transported eastwards with time are observed indicating

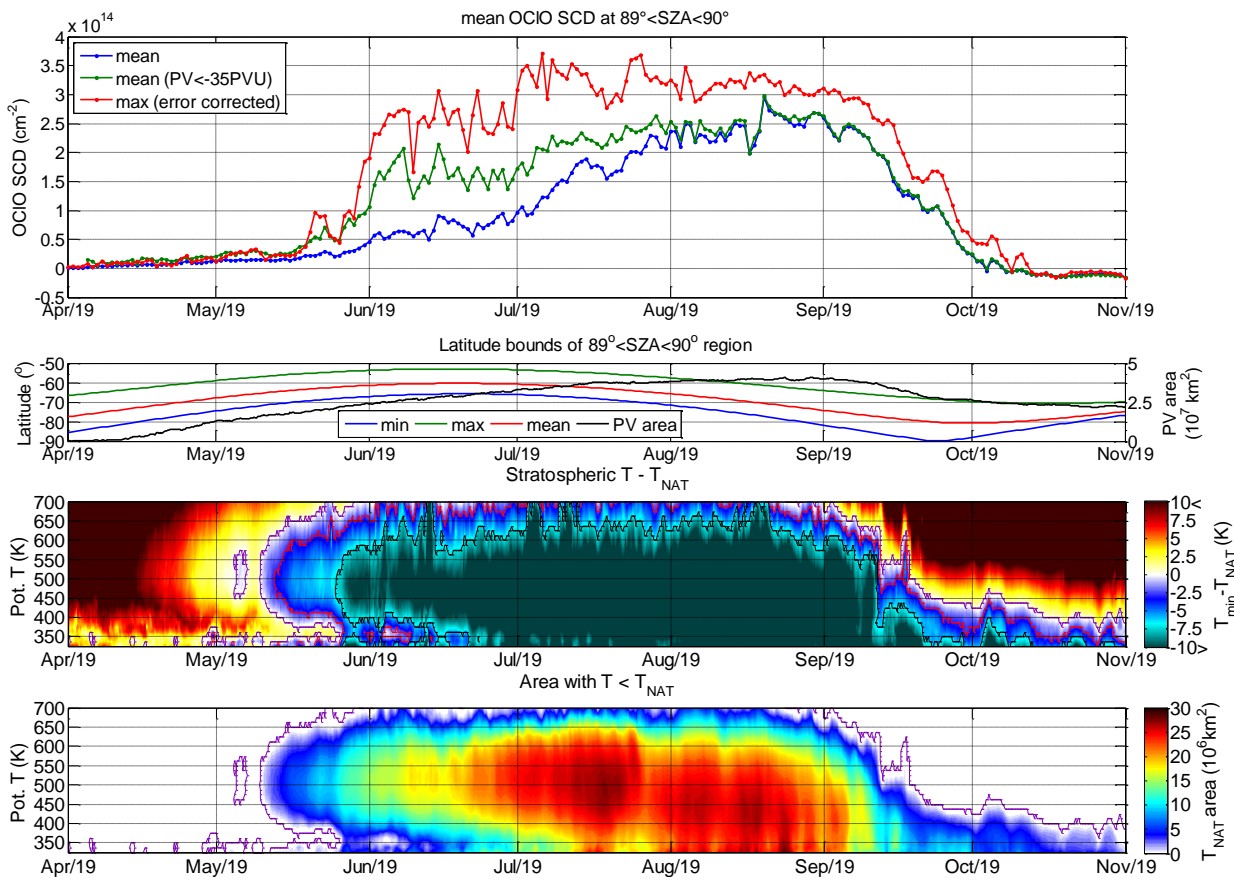

**Figure 13.** Same as Fig. 1 but for the Antarctic winter 2019.

chlorine activation induced by a possible PSC formation due to mountain wave activity. A more consistent PSC evidence in these trails is observed starting in the middle of June. The number of local PSC evidences increases during July and in August for almost all collocated OClO SCDs observations.

Increased OClO SCDs is, as expected, limited to air masses with higher PV (i.e. well inside the polar vortex). The exact PV value above which the OClO SCDs are increased changes during the season: in May high OClO SCDs appear for PV above 40 PVU (it is cold enough for chlorine activation only in the more central parts of the polar vortex). In July the limit decreases to 35 PVU (as the stratosphere cools down also for air masses with lower PV values). Later this boundary increases again along with a strengthening of the polar vortex which is attributed to rising temperatures for given PV values. It is worth mentioning that this strengthening of the polar vortex in late winter and spring in the Southern Hemisphere (SH) has been attributed to a coincidental seasonal temperature increase in the subtropics (Zuev and Savelieva, 2019) which keeps zonal temperature gradients large sustaining the development of the polar vortex. The maximum OClO SCDs increase till the end of June and mostly stay constant during July. At the beginning of September the maximum OClO SCDs begin slightly to decrease

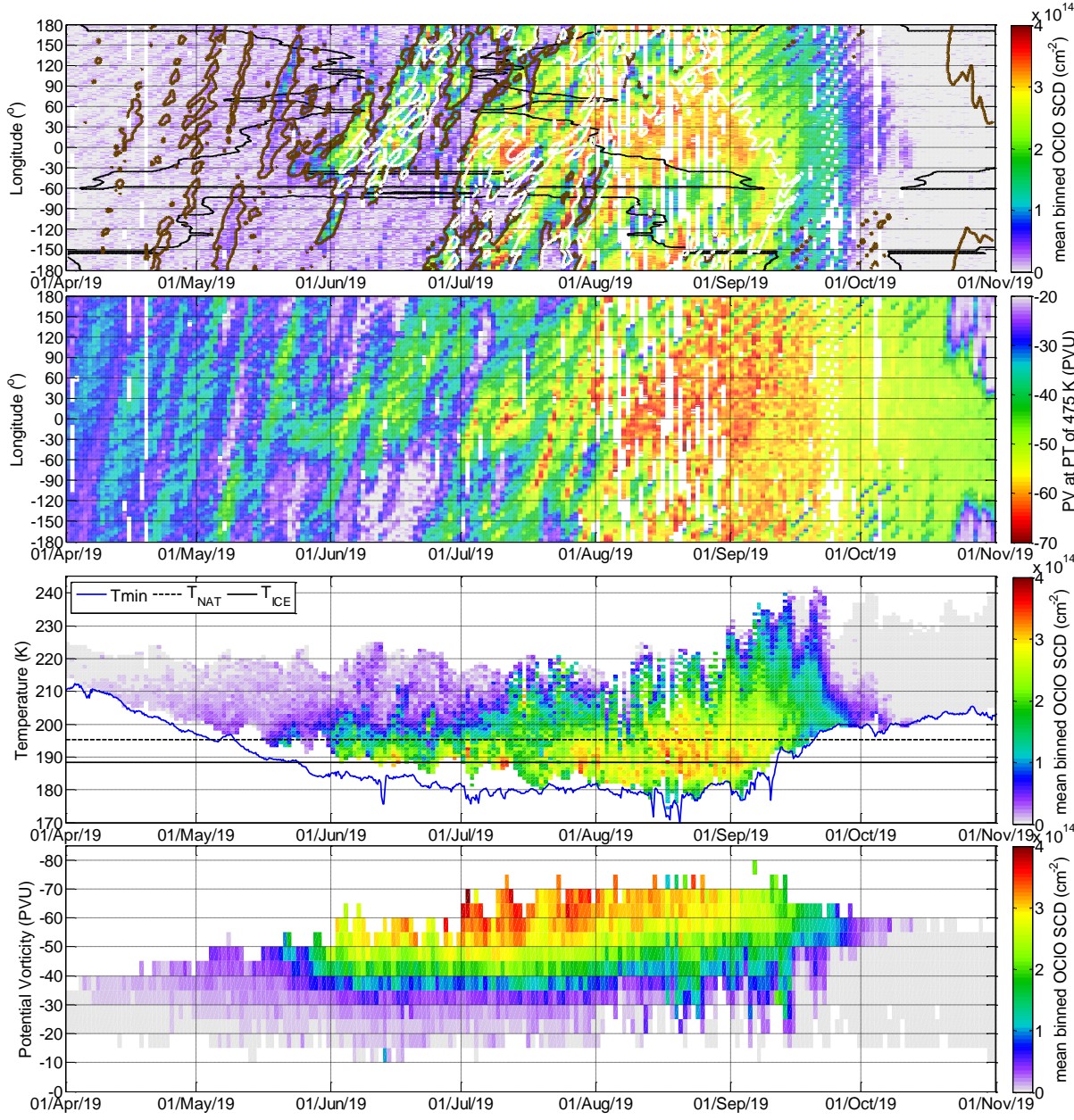

**Figure 14.** Same as Fig. 11 but for the Antarctic winter 2019.

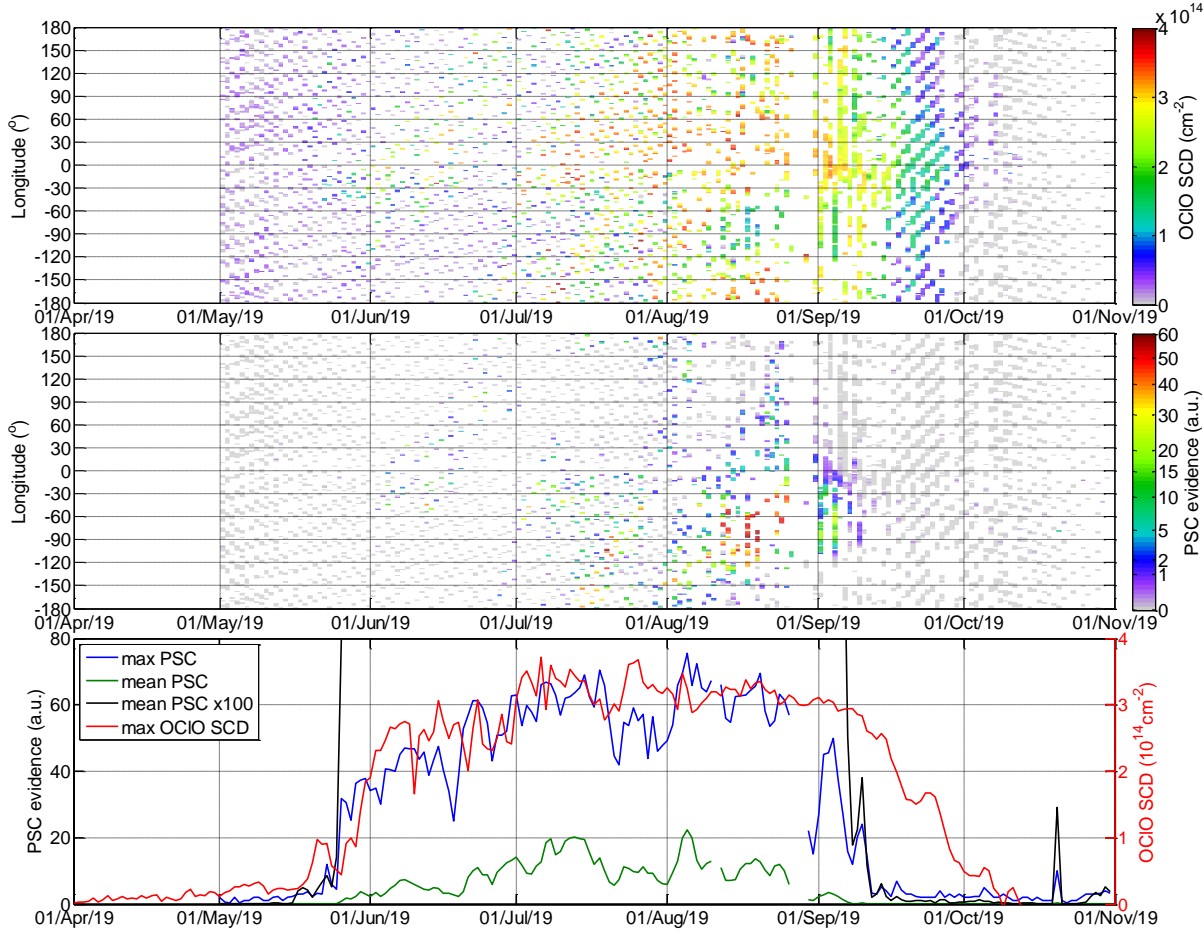

**Figure 15.** Same as Fig. 3 but for the Antarctic winter 2019.

but stay at rather high levels until the last week of September indicating that ClO levels are high enough to enable an effective catalytic ozone destruction. The mean OClO SCDs increase a bit slower till the end of July, which can be explained by the fact that the relationship between PV and the OClO SCDs varies with time and that different areas of the polar vortex (boundary) are observed. Finally, at the end of September to the beginning of October a rather quick chlorine deactivation occurs despite the fact that the temperatures are still below $T_{NAT}$ and the polar vortex is stable. Besides a relation with the decrease in PSCs evidence as observed by CALIOP (or at least PSCs descending to lower altitudes not covered by the considered altitude range of >4 km above the tropopause) at the end of September, also the mechanism of chlorine deactivation as described by Grooß et al. (2011) can play a role: when an almost complete destruction of ozone occurs, almost all chlorine becomes bound in HCl and cannot be reactivated.

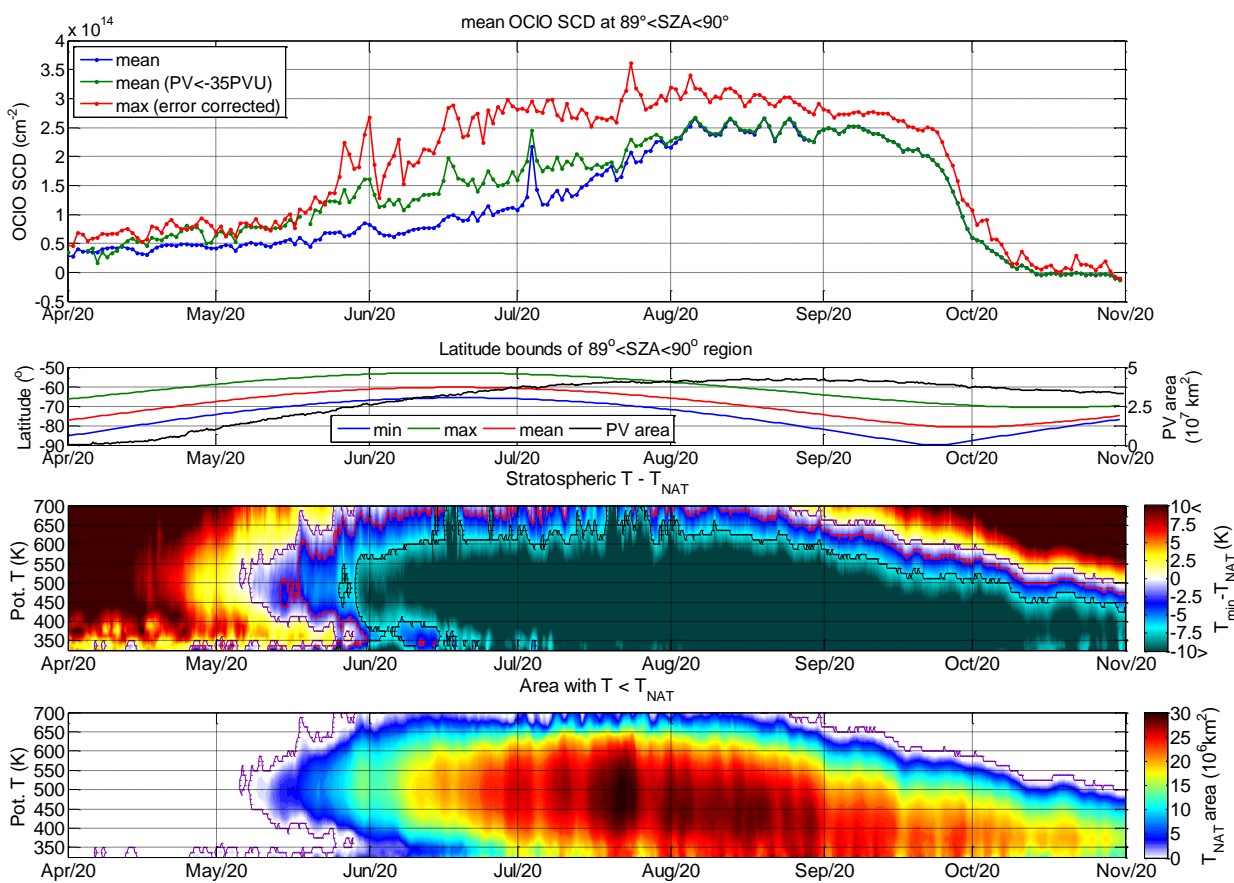

**Figure 16.** Same as Fig. 1 but for the Antarctic winter 2020.

### 4.2.2 Winter 2019

The winter 2019, however, was quite unique as a minor sudden stratospheric warming was observed, which was just a bit weaker than the major sudden stratospheric warming in 2002 (Lee, 2020; Klekociuk et al., 2021). Also a very small ozone hole area in September in comparison to that of 2018 has been reported, but the magnitude of the vortex-averaged chemical ozone depletion was not significantly different between both years. Wargan et al. (2020) attributed most of the smaller ozone loss to dynamics. This is in accordance to Sinnhuber et al. (2003) who reached similar conclusions with respect to the major stratospheric warming in 2002.

The daily mean and maximum OClO SCDs (see Fig. 13) show a similar temporal development as in 2018 until 6 September. Also clearly increased OClO SCDs at local temperatures below $T_{NAT}$ (middle May) and even more increased OClO SCDs at local temperatures below $T'_{NAT}$ (from the beginning of June) are observed (Fig. 14). From beginning of June also evidence for PSCs at the locations with increased OClO SCDs are consistently observed (Fig. 15). After the stratospheric warming (6 September), the area with temperature below $T_{NAT}$ decreases rapidly and the hemispheric minimum temperature rises above

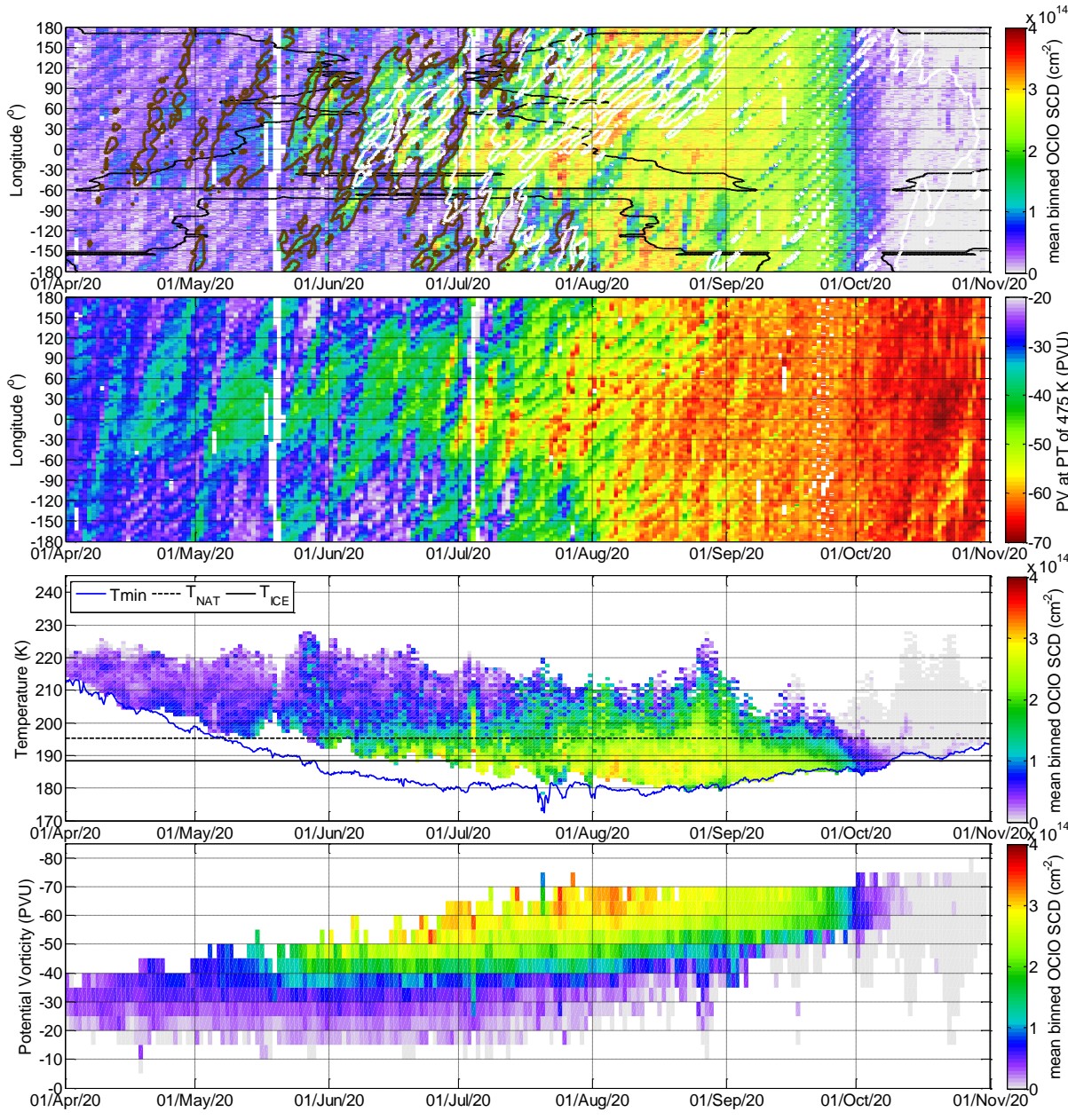

**Figure 17.** Same as Fig. 11 but for the Antarctic winter 2020.

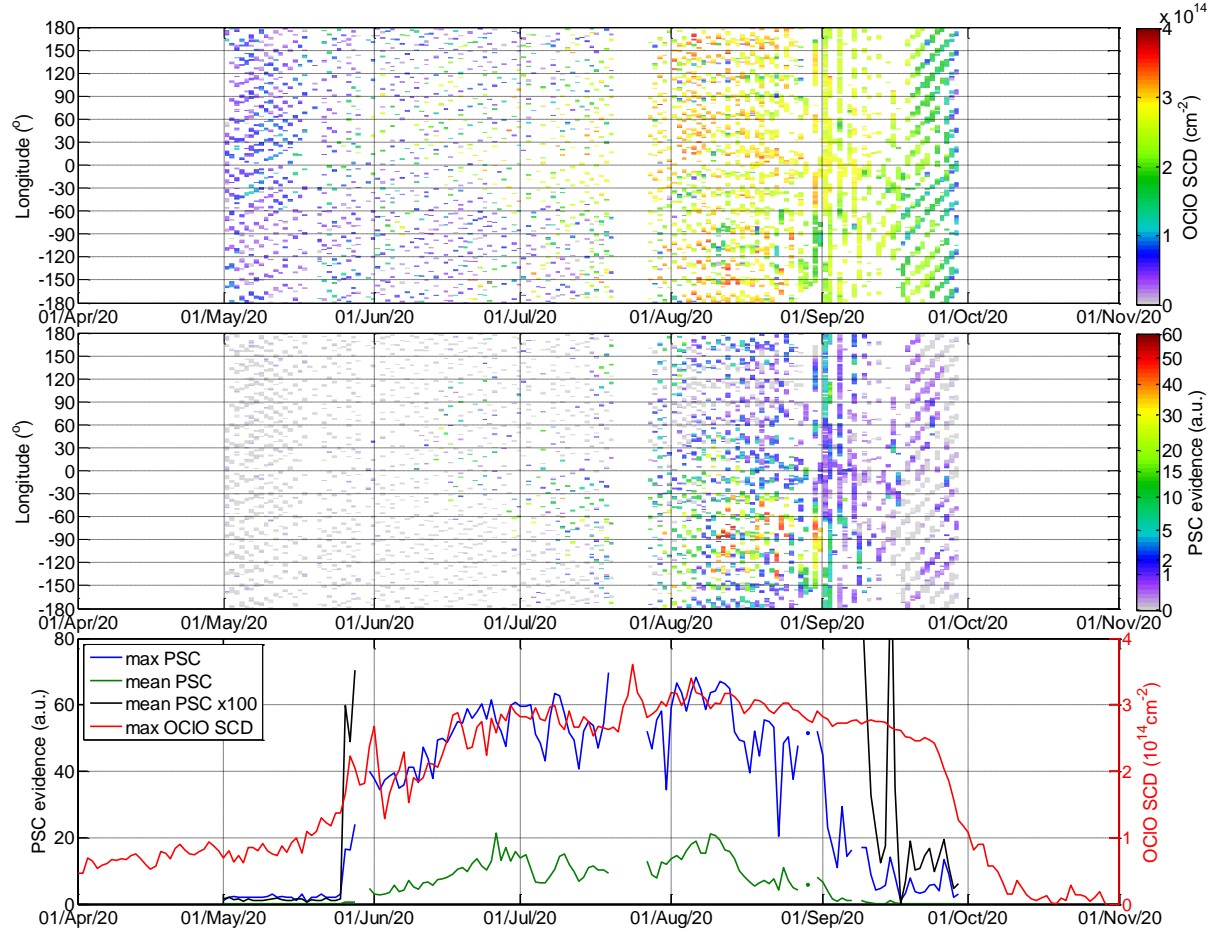

**Figure 18.** Same as Fig. 3 but for the Antarctic winter 2020.

$T_{NAT}$ (at PT 475 K) by the end of the third week of September. The decrease and the rise are accompanied by a strong decrease of the OClO SCDs with a rather constant rate till the end of September. After 6 September also the PSC evidence (both local, as well as the polar mean and maximum) observed by CALIOP becomes almost zero. At the beginning of October the OClO SCDs decrease further at a lower rate. Interestingly, two distinct temperature drops at lower altitudes (at PT around 400 K) lead to two small short-term increases in the mean and maximum OClO SCDs.

Looking on the parameter (longitude, temperature and PT) resolved time series (Fig. 14) one can notice that the high OClO SCDs appear at rather high local temperatures and low PV values already on 11 August and more clearly on several days after 18 August. Also a mixing towards low PV values after 5 September can be seen, being especially strong at the beginning of the second week of this month which coincides with the sudden stratospheric warming episode. The small chlorine activation events at the beginning of October can be seen well distinguished in all parameter resolved time series of OClO SCDs occurring at the lowermost temperatures and the highest PV values. We can speculate that this potential for a further chlorine activation

indicates that not all ozone in the polar vortex was destroyed by the initially activated chlorine. This indicates that chlorine could in principle be reactivated again if the temperatures become low enough, as it is usually the case in the Arctic.

### 4.2.3 Winter 2020

While so far no scientifically peer reviewed analysis of this winter could be found, the SH winter 2020, although with a usual development at beginning, has been reported by meteorolgal surveys (e.g. Copernicus, 2021) with one of the largest, deepest and long persisting ozone holes of the past 40 years during the time period October to December. The earlier months of this winter however shows a vortex development which corresponds to typical Antarctic conditions. Nevertheless a rather similar timing and levels of OClO SCDs and PSC evidences as for 2018 for August to October, thus also during the deactivation period (Figs. 16, 17 and 18) are observed. During June to August, lower OClO SCDs are observed at the coldest temperatures and at highest potential vortex values for this winter than in 2018. An exception however are the already slightly increased OClO SCDs in April (already since mid of March, not shown here). So far we do not have a clear explanation for this finding except of increased backscatter ratios in CALIOP data in May 2020 compared to those in previous years. For the polar mean PSC evidence (black line in Fig. 18, bottom panel) values distinguishable from zero can be observed already at the beginning of May which was not the case for the previous SH winters. The local PSC evidences (Fig. 18, middle panel) have sporadic values slightly above zero which however seem not to be correlated with the collocated SCDs (top panel). Also we do not see a clear local correlation between the backscatter ratios and OClO SCDs when they are at low levels (see Appendix B). The meteorological conditions plotted in Fig. 16 seem to be similar as for the years before with temperature well above $T_{NAT}$. At the beginning of April, the spatial distribution of the increased OClO SCDs is also not associated with areas of high PV within polar vortex (Fig. 17, bottom panel). The OClO SCDs decrease to zero again in October as for the years before, largely excluding the possibility of a systematic instrumental effect. Note that a similar increase is also consistently observed in the preliminary Sentinel5P Innovation activity (S5p+I) operational TROPOMI OClO product (Mayer et al., 2020) OClO SCD data and the ground-based zenith sky observations at Neumayer station in Antarctica show a slightly larger diurnal variability in April and May than for the previous two winters as shown in Puķīte et al. (2021).

## 5 Conclusions

We related our new dataset of TROPOMI OClO SCDs to meteorological parameters driving polar vortex dynamics and thus also PSC formation and chlorine activation. OClO SCDs are also compared directly to PSC measurements from CALIOP on CALIPSO. The great advantage of satellite observations was exploited in the way that, in addition to the temporal evolution of the chlorine activation, also its spatial features were investigated. The TROPOMI OClO SCDs are generally well correlated with meteorological parameters. The most important findings are: The chlorine activation signal appears as a sharp gradient of the OClO SCDs once the local temperature drops approximately below $T'_{NAT}$ (3 K below $T_{NAT}$) thus beeing in agreement with previous research. For the NH the sharp increase is also well related to such a dropping of the hemispheric minimum temperature (possibly because of a better mixing of air masses within the vortex) while in the SH a weaker relation with

respect to the hemispheric minimum temperature is found. Also a relation with the lee sides of mountains can be observed at the beginning of the winters indicating a possible association of OClO formation to lee waves.

The comparison of the OClO SCDs to PSC measurements from CALIOP on CALIPSO reveals that increased OClO SCDs in most instances coincide well with CALIOP measurements where PSCs are detected. Increased OClO SCDs however do not always coincide with enhanced PSC evidence. While in many cases increased OClO SCDs without coinciding PSC could be caused by transport or mixing and the presence of PSCs somewhere else in the polar region, at the beginning of winter the observed moderate levels OClO SCDs could not be clearly associated with a PSCs presence detected by CALIOP.

High OClO SCDs reaching at maximum $3 \times 10^{14}$ cm$^{-2}$ are observed for the very cold stratospheric NH winter 2019/2020 with its very stable polar vortex thus being close to the maximum values found for the SH winters.

An extraordinary winter in 2019 in the SH was observed with a minor sudden stratospheric warming at the beginning of September. Until this event similar OClO SCDs in this winter were observed compared to the previous winters, but the deactivation occurred about $1-2$ weeks earlier in this winter.

Further investigation are still needed with respect to the exceptional OClO increase which goes along with increased backscatter ratios compared to previous winters but is not correlating with the stratospheric meteorology in late March and April in 2020 in the SH where a larger OClO SCD signal above the typical uncertainty range was observed ($\sim 5 \times 10^{13}$ cm$^{-2}$) which is also observed in the S5P+I data.

*Data availability.* Data are available upon request

## Appendix A: Radiative transfer effects on the sensitivity area of OClO SCDs

At high SZAs direct sunlight crosses the atmosphere at very slant paths. Afterwards (with or without undergoing additional scattering before) it is scattered along the line of sight towards the instrument. The distribution of the light that is detected by the instrument is expected to vary both vertically and horizontally depending on the scattering and absorbing properties of the atmosphere and the ground (e.g. the air density, trace gas concentration or PSCs presence, ground albedo), of the light (i.e. wavelength) and the solar and viewing geometries. OClO slant column densities (SCDs) can most directly be interpreted as OClO number densities integrated along the light paths that contribute to the measurement. Thus the contribution of a certain area to the measurement depends both on the light paths that cross this area and the OClO number density there.

We use the 3D full spherical radiative transfer model (RTM) McArtim (Deutschmann et al., 2011; Deutschmann, 2014) to quantify the spatial sensitivity of the measured OClO SCDs by obtaining so called box AMFs $B_i$:

$$B_i = \frac{L_i}{\Delta h_i} \tag{A1}$$

where $L_i$ is the effective light path in the 2D box $i$ with a vertical resolution $\Delta h_i$, also being horizontally resolved along the the direction from the line of sight coordinate towards the Sun.

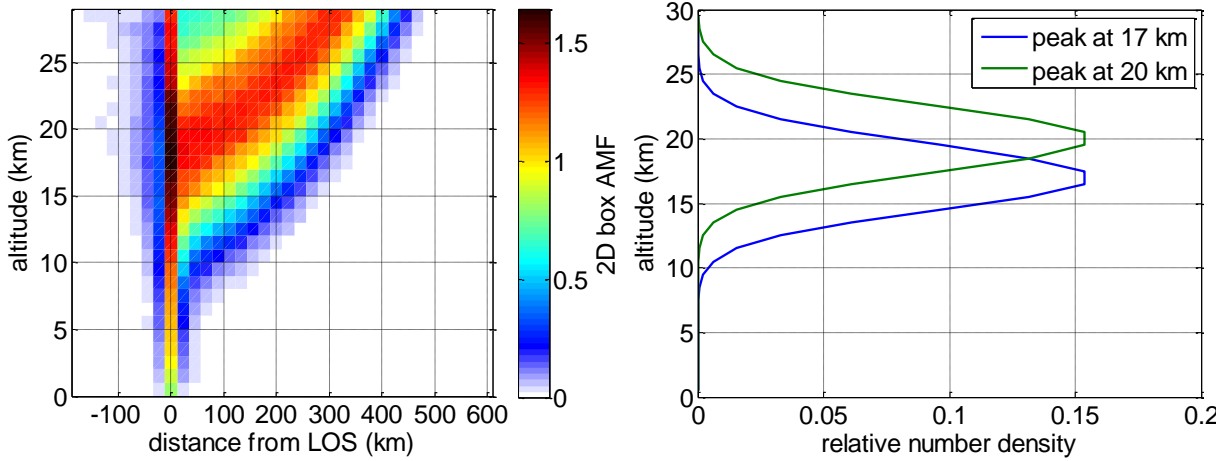

**Figure A1.** Left: 2D box AMFs for a clear sky atmosphere. Right: Relative OClO number density profiles used for the sensitivity studies.

Box AMFs obtained for an aerosol free atmosphere (with a 1 km vertical resolution and a 0.2° ( 22 km) horizontal resolution) at a SZA of 89.5° at the measurement location are illustrated in the left part of Fig. A1. As expected, the largest box AMFs (effective light paths) occur near the line of sight position at altitudes of around 19 km. Also areas where the light has travelled through have increased box AMF values.

To evaluate the contribution of these areas to the OClO SCDs we need to multiply these box AMFs with the local OClO number densities. To consider the variability in the OClO distributions we base our calculations on OClO number density profiles which are shifted in altitude. We consider here two Gaussian shape profiles with FWHM of 6 km with peaks at 17 or 20 km. The profiles are illustrated in the right panel of Fig. A1. With regard to the horizontal variability we assume that the OClO photolysis rate varies linearly from $0.015°$ s$^{-1}$ to $0.05°$ s$^{-1}$ between SZAs of 90° and 85°, consistent with the calculations in

Kühl et al. (2004a), thus OClO decreases in the direction of the Sun proportionally to the inverse of the photolysis rate. Note that atmospheric dynamics could increase the gradient even further (as typically potential vorticity decreases towards lower latitudes), so the evaluation here would represent the largest horizontal extention scenario.

The left plots in Fig. A2 show the obtained partial SCDs (products of box AMFs and local OClO number densities) calculated for the profile with the peak at 17 km (top) and 20 km (bottom), respectively. Horizontally and vertically integrated SCDs are

470 shown on the right (top and bottom plots, respectively). A clear maximum at the line of sight location is obtained at altitudes slightly above the simulated profile peak with an excentric distribution towards the direction of the Sun. We find that the mass centre of the sensitivity area for this simulation is located 100 km (for the profile with peak at 20 km) and 80 km (peak at 17 km) from the line of sight coordinate.

The possible presence of PSCs in the stratosphere are supposed to provide a strong effect on the scattering properties and

475 thus also to affect the stratospheric OClO measurements. Therefore, we have performed tests with variable PSC amounts based on the aerosol climatology as presented in Vanhellemont et al. (2005) inside the polar vortex. We found that the mass centre for cases with PSCs present are now slightly closer to the line of sight location. Assuming PSC profiles with a three times larger

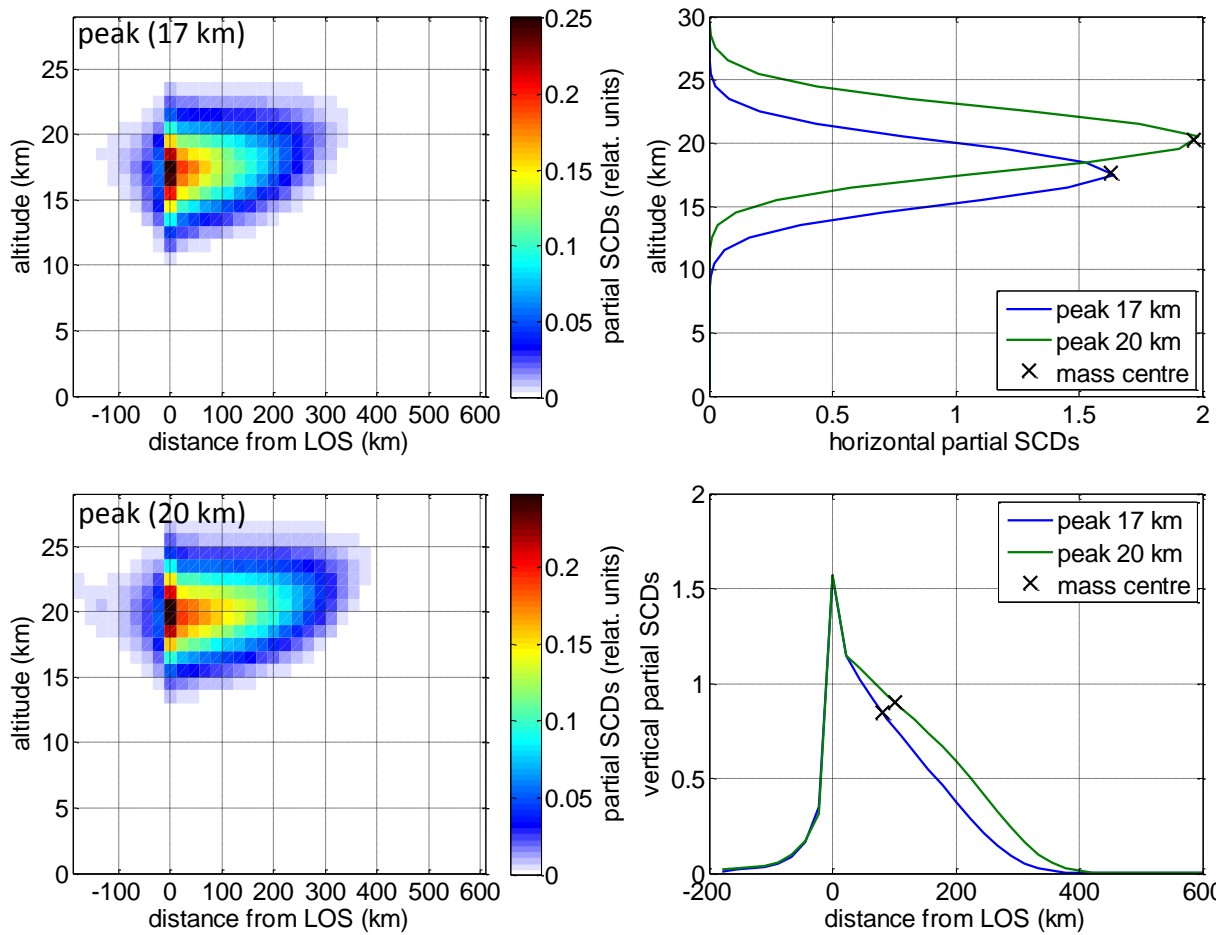

**Figure A2.** Left: Simulated partial SCDs (products of the box AMFs and the local OClO number density values) calculated for the profile with the peak at 17 km (top) and 20 km (bottom). Right: Horizontally (top) and vertically (bottom) integrated partial SCDs with mass centres indicated.

PSC extinction than the presented median values in the climatology, the mass centre of the sensitivity area is located 60 – 80 km (depending on the assumed peak altitude) from the line of sight coordinate. Thus the effect is rather limited.

To test the effect on the comparison plots we considered as the measurement location the coordinate that is 100 km towards the Sun from the line of sight coordinate, i.e. the maximum possible displacement of the mass centre of the sensitivity distribution. We repeated the calculations with this assumption for the winter 2019/2020 in the NH and 2020 in the SH. The plots are shown in Figs. A3 and A4. The results for the shifted measurement coordinate follow very well those with the considered measurement location at the line of sight coordinate (as in Figs. 8 and 17). Only a shift by about 1 K temperature between both

results is observed. The shift in PV is somehow larger near the polar vortex boundaries (due to the larger gradient) but is still below the resolution of the PV used in the plots (5 PVU).

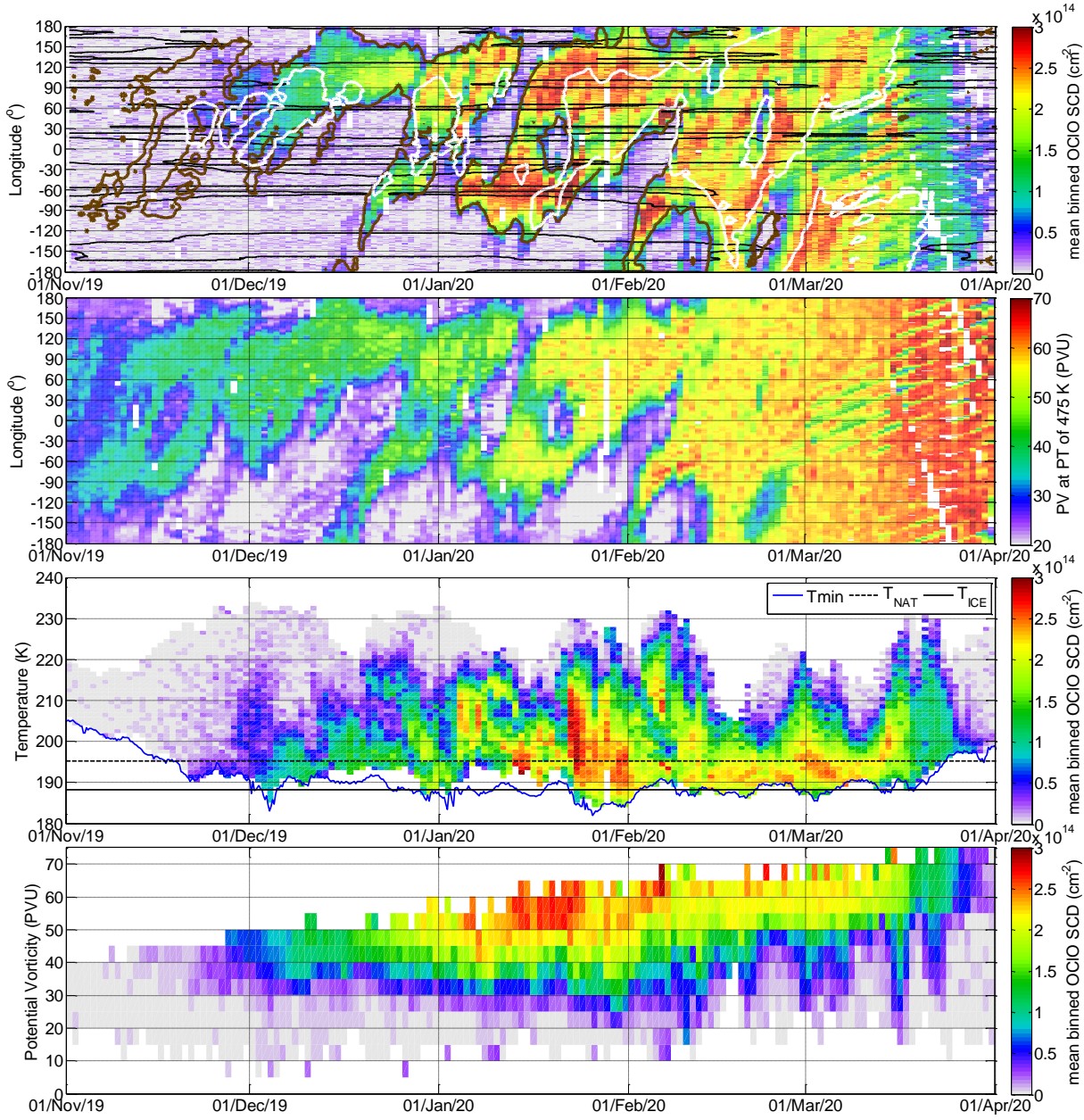

**Figure A3.** Same as Fig. 8 but with the assumed measurement location coordinate shifted by 100 km from the line of sight coordinate towards the Sun.

Also for the comparison to the PSC evidence the shift of 100 km towards the Sun has no substantial effect (compare Fig. A5 for the winter 2020 in SH with the corresponding plots in Fig. 18) despite the lower PSC evidence values in the first half of the winter.

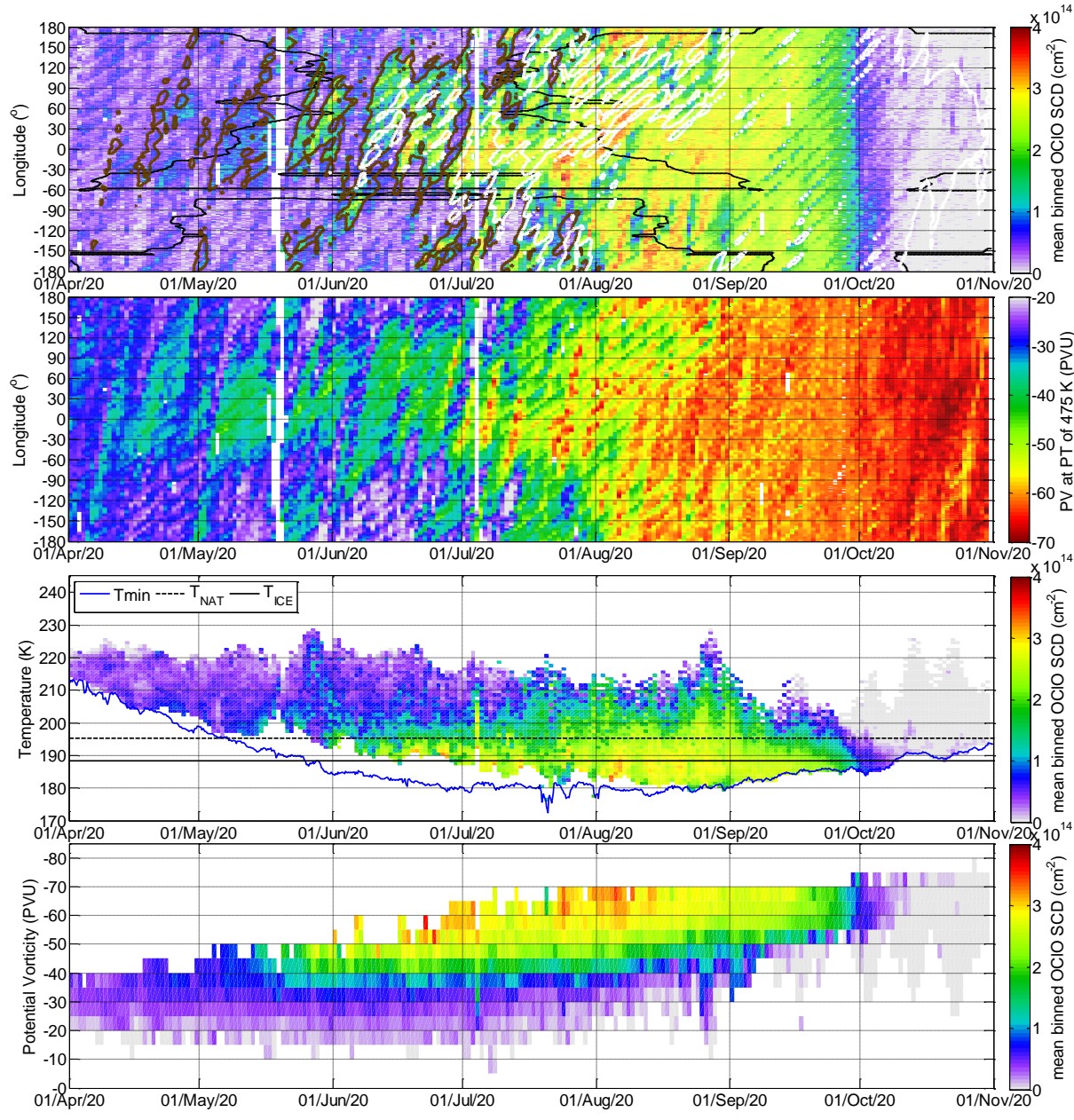

**Figure A4.** Same as Fig. 17 but with the assumed measurement location coordinate shifted by 100 km from the line of sight coordinate towards the Sun.

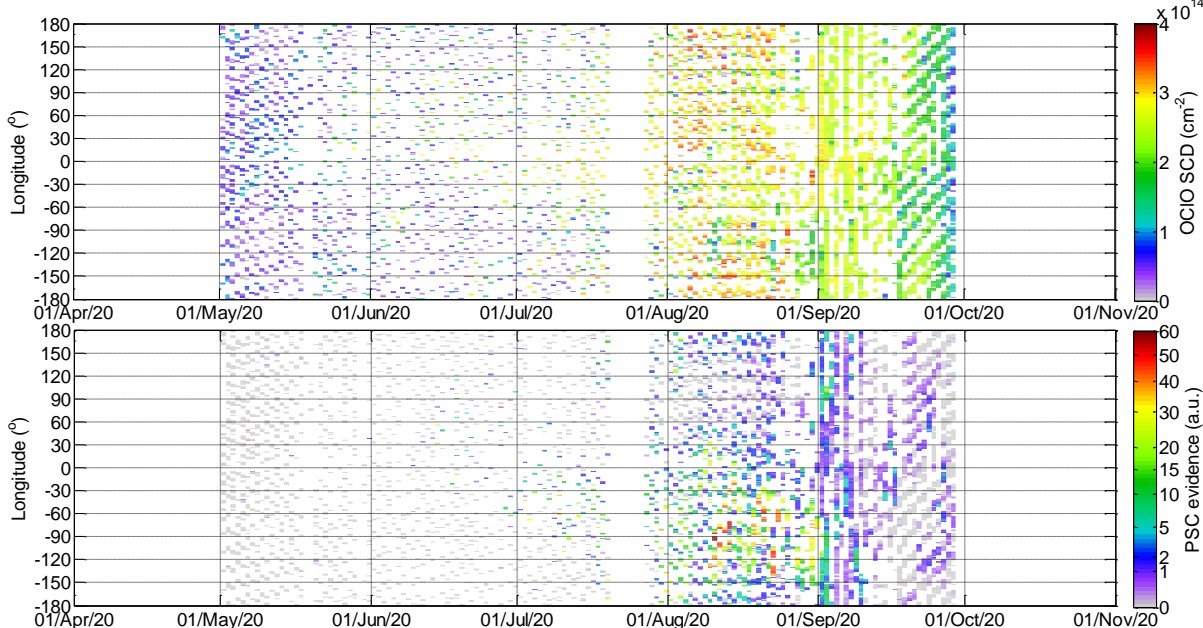

**Figure A5.** Same as the two upper plots in Fig. 18 but with the assumed measurement location coordinate shifted by 100 km from the line of sight coordinate towards the Sun.

## Appendix B: PSC evidence and aerosol backscatter ratio in comparison to OClO SCDs

Here, we investigate how well the PSC evidence can represent PSCs in comparison to the aerosol backscatter ratios for the comparison with the OClO SCDs in a case study for months during the PSC activation period (May-July 2020 in SH). Therefore, the mean backscattered ratio for the same altitude range (i.e. for altitudes above 4 km above the tropopause) and for the same collocation criteria as for the PSC evidence (Sect. 3) was calculated from the CALIOP data. The OClO SCDs, and the mean backscatter ratios are plotted in Fig. B1. In the top panel of Fig. B2, the correlation plot between the PSC evidence and the mean backscatter ratios are shown, left for May-July 2020, right only for May 2020. Please note the different x- and y-axis scales scales. It is found that in the case of low mean PSC backscatter values (in May) no good correlation with the PSC evidence can be seen. Since PSCs are detected by testing whether the total backscatter ratio and/or the perpendicular backscatter are above a certain threshold at individual altitudes (Pitts et al., 2009), it is understandable that for the mean backscatter ratios there is no clear threshold. By including also time periods with large PSC backscatter values, however a more clear relationship can be seen. Only few cases with very large mean backscatter ratios fall out of the slope because the representation of the PSC abundance by the PSC evidence is limited by the different spatial averagaring intervals, with the minimum being the spatial resolution of the individual measurement. Thus the PSC evidences for very dense PSC clouds are underestimated.

In order to understand which quantity better relates with the OClO SCDs, we plot the correlation between the OClO SCDs and either the PSC evidence or the mean backscatter ratio (middle and bottom panel of Fig. B2, respectively). In the case of

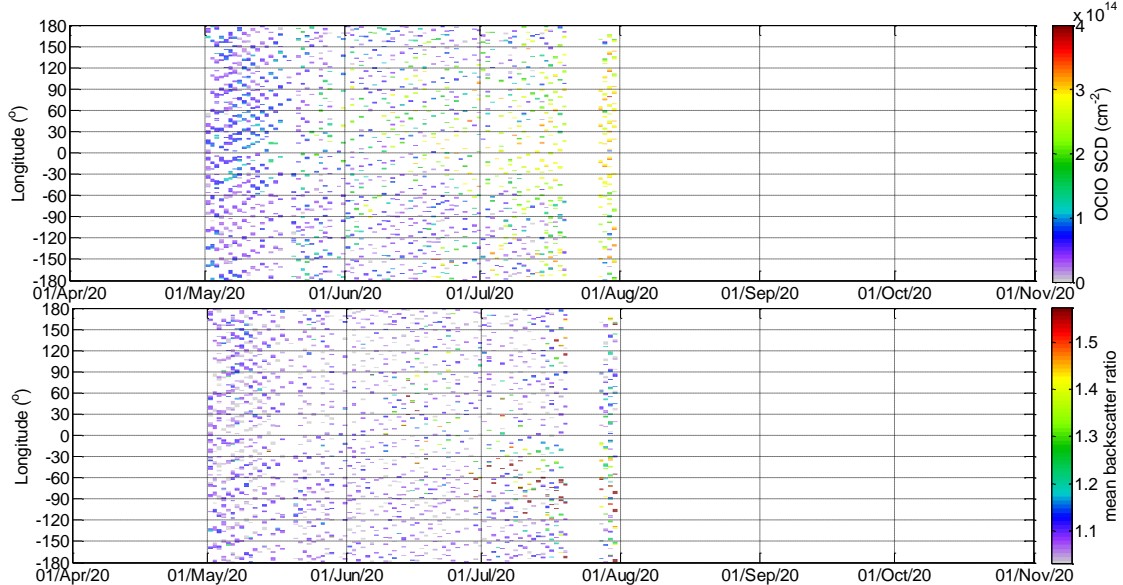

**Figure B1.** Upper panel: same as the upper panel of Fig. 18 but with data for May – July 2020. Bottom panel: collocated mean backscatter ratios.

low PSC/stratospheric aerosol levels (especially in the plots for May 2020) there is a better relation between the OClO SCDs and the PSC evidence: in case of an increased PSC evidence, OClO SCDs are generally increased (distinct from zero). For the mean backscatter ratio, the scatter of the data points is too large to see a relation between this quantity and the OClO SCDs for May. Given the somewhat better sensitivity of the PSC evidence for low PSC levels, we conclude that the PSC evidence is a
510 well suited quantity for the detection of PSCs from CALIOP.

*Author contributions.* J.P. with support of C.B. S.D. M.G. and T.W. performed the study and analysed the results. C.B. with support of J.P. and T.W. retrieved OClO SCDs from TROPOMI measurements. S.D. downloaded and maintained the local ECMWF dataset. J.P. prepared the manuscript with supervision by T.W and comments by all co-authors.

*Competing interests.* No competing interests are present

*Acknowledgements.* We acknowledge ESA and SP5/TROPOMI team for providing TROPOMI L1b data. We acknowledge the use of ECMWF ERA5 data: we use the modified Climate Change Service information and/or modified Copernicus Atmosphere Monitoring Service information (for the years 2017-2020). Neither the European Commission nor ECMWF is responsible for any use that may be made of the

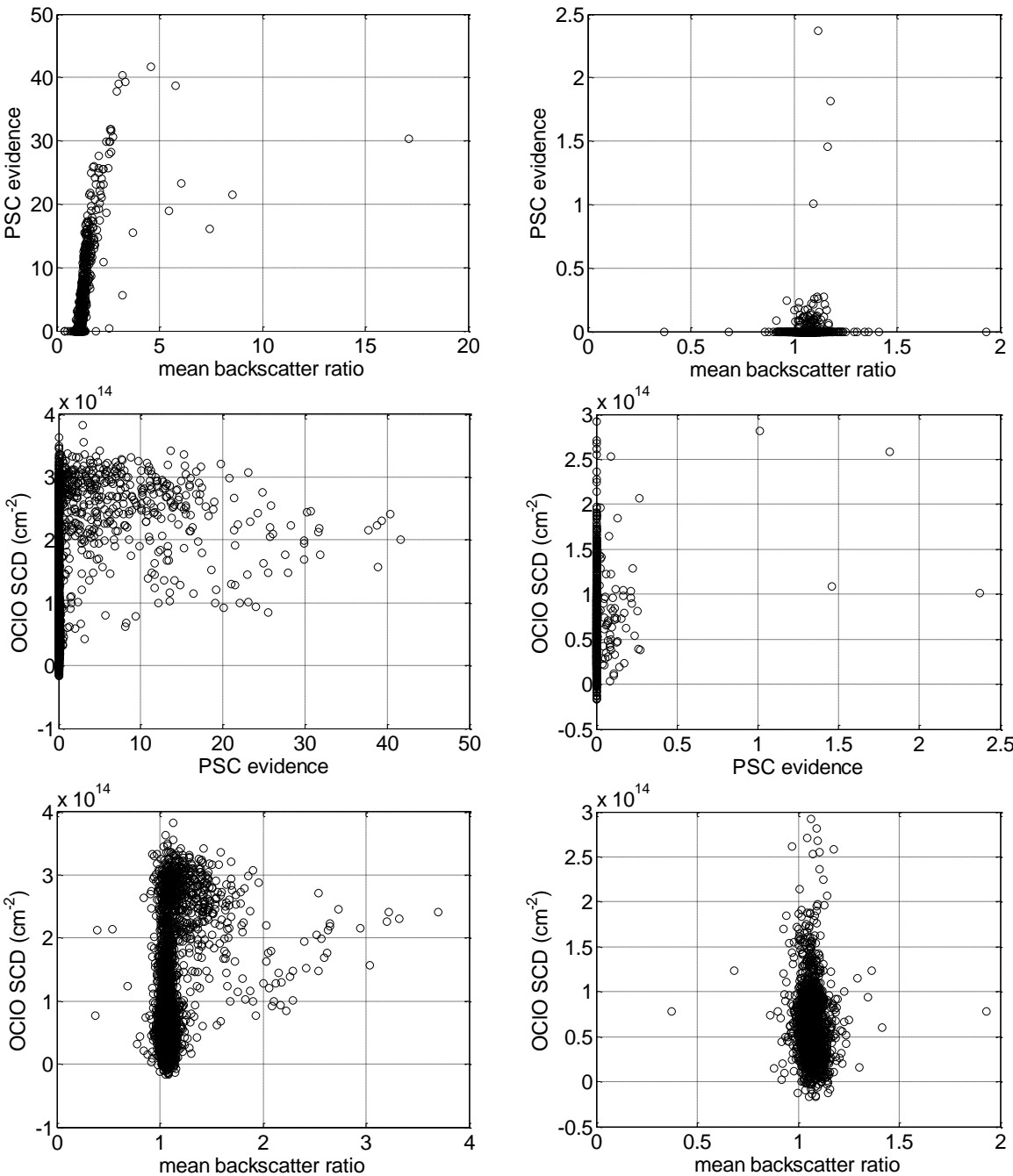

**Figure B2.** Upper panel: Correlation plots between PSC evidence and the mean backscatter ratio. Middle panel: Correlation plots between the OClO SCDs and the PSC evidence. Bottom panel: Correlation plots between the OClO SCDs and the mean backscatter ratio. On the left side, data for May-July 2020 are considered, on the right just for May 2020.

Copernicus information or data it contains. We also acknowledge NASA and CALIPSO/CALIOP team for the Cloud-Aerosol Lidar and Infrared Pathfinder Satellite Observations (CALIPSO) Lidar Level 2 Polar Stratospheric Clouds (PSC) Mask, Provisional Version 1-10 data product. These data were obtained from the NASA Langley Research Center Atmospheric Science Data Center. Last but not least we thank Udo Frieß, Carl-Fredrik Enell, Uwe Raffalksi and Andreas Richter for their fruitful comments to this paper and their contributions to the validation of the new OClO data set as presented in Puķīte et al. (2021).

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
