# Peer review of "OCIO as observed by TROPOMI: a comparison with meteorological parameters and PSC observations"

_Atmospheric Chemistry and Physics, 2021_

## Author Comment (AC1)

Response to the Reviewer #1

We thank the Reviewer for the constructive review and address the comments below.

General Comments:

In this work, Differential optical absorption spectroscopy (DOAS) technique is applied to TROPOMI data to obtain OClO Slant column densities (SCDs), for Arctic and Antarctic latitudes, from November 2017 until October 2020. These SCDs have been also compared with meteorological data from the ECMWF model (temperature and potential vorticity) and CALIOP PSCs observations. Through this study, the temporal and spatial evolution of the OClO SCDs can be examined, as well as the correlation with the studied parameters, allowing also identifying possible causes of chlorine activation. A comparison between both hemispheres has also been presented, and some interesting unusual episodes concerning formation, development or deactivation of polar vortex have been studied.

The research performed in this work has been clearly presented and explained and represents useful information for the Atmospheric science community. Thus, I think that this paper should be published in ACP. However, I think that some questions should be clarified.

Specific Comments

- Has some cloud-screening been applied to the DOAS data? Could tropospheric clouds have a significate impact in the presented DOAS measurements?

No cloud screening has been applied. Since OClO as a stratospheric trace gas is above the tropospheric clouds, no cloud shielding occurs. There can still be a small effect on the air mass factor due to the dependency of multiple scattering effects on the backscatter albedo (up to 5-10%) which, however, certainly would not justify a cloud filtering.

We add to the manuscript at the end of the paragraph about L95: "Furthermore, the occurrence of OClO in the stratosphere ensures that no cloud filtering needs to be applied because no shielding by tropospheric clouds is expected."

- Page 5, lines 133-135: Most of the information provided by the DOAS measurements come from air mases located at certain altitude and distance from the observation point, depending on the geometry of observation, Solar zenith angle, etc.. Has been this taken into account in the comparison between the TROPOMI and the ECMWF or CALIPSO data? Is this what you mean when talking about the multilinear interpolation? Do you use a spherical radiative transfer model to do so?

The described collocation procedure considers the instrument viewing geometry by interpolating the meteorological data to the geographic coordinate along the instrument's line of sight at 19.5 km (as already stated in the paper). The multilinear interpolation means a trilinear interpolation of the meteorological parameters to this coordinate (latitude, longitude) as well as the time of the measurement. To make it more clear, we replace "multilinear" by "trilinear" in the manuscript. The consideration of radiative transfer would necessarily require

a-priori constraints about the concentration variability along the light path which, given the high spatial variability of the OClO number density, would mean a dependence on additional constraints on the atmospheric state like chemical composition and PSC distribution which would introduce additional uncertainties. Thus no radiative transfer modelling is applied in these calculations. In response to the comment of the reviewer and also given that such an investigation up to our knowledge has not been done so far, we performed a sensitivity study by means of a 3D radiative transfer model to estimate the range of the possible sensitivity area of the OClO SCDs measurements. Also the possible effect of a horizontal shift of the comparison location towards the Sun is investigated. We found that the effect on the comparison is rather limited thus not affecting the findings of the manuscript.

We added the following statement about these findings to the paper:

"No radiative transfer modelling is applied during the assignment. Radiative transfer effects indicate that the mass centre of the sensitivity area of the measured OClO SCDs is expected to be located towards the direction of the Sun from the line of sight coordinate. The consideration of the radiative transfer would require a-priori constraints about the spatial variability of the OClO number density. Given its high variability and also the dependence of RTM on additional constraints on the atmospheric state, especially also the highly variable PSC distribution, it would introduce additional uncertainties. We have found in sensitivity studies (see Appendix A) that this displacement is expected to be less than 100 km and typical PSC concentrations do not largely affect it. It is thus below the resolution of the applied meteorological data set and the systematic effect on the performed comparison is estimated as rather limited (variation in temperature of 1K and below and in potential vorticity of 5PVU or below), therefore not affecting the findings of the study."

We also provided the details of the investigation in the Appendix A

- Second panel from top of figure 2 and similar figures: Just as suggestion, the colour scale of these colour maps are contrary to the rest of the panels of these figures (red means low values of PV and blue means high values). Perhaps, using similar colours scale for all the panels would be more visually intuitive.

We selected a contrasting colour scale for this panel because it shows a different quantity in contrast to the other panels. But we can follow the suggestion and use the same colour scale if this seems more intuitive.

- Figures using "Longitude" as Y axis: even if positive and negative values of longitude are usually assigned to East and West longitudes, respectively, this should be clarified somewhere in the figure captions or in the text.

We added this clarification in the figure captions.

- Page 12, line 211 and page 13, line 212: The provided longitude values correspond to East longitudes instead West longitudes, Is it right?

Yes, indeed. We corrected this typo.

- Page 16, line 242: The provided OClO SDCs values include also those below the detection limit?

We do not filter the OClO data set in the figures just to show SCDs above the detection limit. Instead we have discussed and provided the detection limit in Sect. 2. We just pay attention here that the observed enhancements during the last days of November are very small (technically below the detection limit) but discuss them since they are persisting for several days (hence they seem statistically significant)

- Page 28, lines 407-409: The commented exceptional OClO increase could be related to aerosols, as commented previously by the authors (page 3, line 59)?

In principle we agree with the reviewer that there could be a relation. Indeed we see increased backscatter ratios in May 2020 comparing to those in previous years. However we do not see a clear local correlation between the backscatter ratios and OClO SCDs when they are at low levels. We added this information to the text by changing the description for the SH winter 2020:

So far we do not have a clear explanation for this finding except of increased backscatter ratios in CALIOP data in May 2020 compared to those in previous years. For the polar mean PSC evidence (..) values distinguishable from zero can be observed already at the beginning of May which was not the case for the previous SH winters. The local PSC evidences (..) have sporadic values slightly above zero which however seem not to be correlated with the collocated SCDs (top panel). Also we do not see a clear local correlation between the backscatter ratios and OClO SCDs when they are at low levels (see Appendix B).

We modified also the last paragraph of the conclusions:

Further investigation are still needed with respect to the exceptional OClO increase which goes along with increased backscatter ratios compared to previous winters but is not correlating with the stratospheric meteorology in late March and April in 2020 in the SH where a larger OClO SCD signal above the typical uncertainty range was observed (5E13 cm^-2) which is also observed in the S5P+I data.

Technical Corrections:

- Some sentences are too long. I think some "," should be introduced. As example: Page 2, lines 29-31; Page 6, line 166: "For the comparison, .."; Page 6, line 169: "In addition, .."; Page 6, line 166: "For this winter, .."; etc.

We proceeded as suggested. We also rely on the English proofreading service offered by the Copernicus office.

- Page 4, line 113: Introduce the meaning of the ECMWF acronym

We introduced now the meaning at the first occurrence (same page, line 99)

- Page 5, line 135: "..19.5 km of altitude".
- Page 5, line 137: "..The obtained correlative dataset..".

All corrected as suggested

---

## Author Comment (AC2)

We thank the Reviewer for the constructive review and address the comments below.

In this paper, the new TROPOMI OClO slant column density (SCD) product developed by the MPIC group is compared to meteorological data for both Antarctic and Arctic regions for the first three winters of the S-5p satellite mission (November 2017–October 2020). A good qualitative correlation is generally obtained in both hemispheres between the OClO SCD and the selected meteorological parameters, namely the minimum polar hemispheric temperature, the polar vortex area, and the area where air temperature is below the temperature of nitric acid trihydrate (NAT) PSC particles formation. In addition, the TROPOMI OClO SCDs are also found to coincide well with PSC observations from the CALIPSO Cloud-Aerosol Lidar with Orthogonal Polarization (CALIOP) PSC observations. The various high OClO level periods observed in both Northern and Southern polar winters are discussed in terms of polar vortex activation and deactivation processes and stability.

This study fits well with the scope of ACP. Moreover, the manuscript is clearly structured and the method and results are generally presented and discussed in an appropriate and balanced way. Therefore I recommend the paper for publication in ACP after addressing the following comments:

General comment: This is a suggestion for a future study rather than a comment to address here but it would be interesting to include also the TROPOMI BrO and $O_3$ column data sets in the loop. Comparing those data sets with the presented OClO and PSC observations and meteorological parameters could provide a unique opportunity to investigate the relationship between halogens activation, stratospheric ozone depletion and meteorological conditions during the last three winters, especially in the Northern polar region where the polar vortex can be highly variable.

Many thanks to the reviewer for this suggestion! We will consider this in further studies.

Specific comments:

Page 2, line 46: Maybe you should give the typical solar zenith angle threshold value above which the OClO abundance can be detected from passive DOAS measurements. A number for the detection limit (in molec/cm2) should be also given here.

This statement is to say that OClO can best be investigated at high SZAs because for such conditions the signal to noise ratio of the retrieved OClO SCDs can become largest. The detection limit and thus the SZA threshold, for which enhanced OClO SCDs might be detected, vary from instrument to instrument. Also different statistical processing like averaging over certain space and time intervals may change it. For TROPOMI we can retrieve OClO down to 65° SZA with a typical detection limit below 2E13cm-2 for a 20x20 km2 area.

We added this information to the manuscript by modifying and expanding the paragraph at line 95:

"The detection limit and thus the SZA threshold, for which enhanced OClO abundances might be detected, vary from instrument to instrument. Further it varies with SZA due to different signal to noise ratio, also different statistical processing like averaging over certain space and time intervals may change it. A detection limit of about 0.5—1x10^14 cm-2 have been estimated at SZA of 90° for SCDs gridded on a resolution of 20x20 km^2 which is well suited for measurements in the stratosphere. We can retrieve OClO slant column densities (SCDs) with a typical detection limit below 2x10^13 cm-2 for the 20x20 km^2 area down to 65° SZA."

Page 4, lines 93-97: Did you apply any filtering on cloudy pixels in the construction of your OClO SCD gridded product? Since the OClO formation is enhanced in the presence of PSCs, how the latter can influence the quality of your OClO retrieval? Please elaborate.

No filtering with cloudy pixels is performed because the effect of clouds is very limited (please see also the answer to Reviewer 1). To retrieve OClO SCDs no input about the atmospheric properties is needed. Above clouds even the signal to noise ratio is typically increased because of more backscattered light, thus the quality (i.e. retrieval error) of the retrieved OClO is even better.

Concerning OClO in the presence of PSCs it is true that the measured OClO SCDs not only depend on the OClO concentration but also on the length of the light path (which can be affected by PSCs). The latter dependency, however, is difficult to quantify for each measurement because of the high atmospheric variability and the missing detailed information about it.

While evaluating the radiative transfer effects concerning the spatial sensitivity (see also the corresponding comment by the reviewer #1), we checked also the effect of PSCs. We found that the PSC effect is limited, and thus still a semi quantitative comparison (as presented in the paper) is meaningful.

We added this information to the text (as formulated in the response to the comment by the reviewer #1) and provided details of the sensitivity study in Appendix A.

Page 4, line 120: The SZA range (89-90°) used for the selection of OClO SCD should be better justified. Did you test other SZA ranges since both the altitude of the air mass probed by the TROPOMI sensor and the altitude of the maximum OClO concentration peak depend on the SZA?

The selected SZA range is motivated by a larger ratio between the OClO SCDs and the detection limit in this range, i.e. the amplitude of the observed OClO SCDs decreases faster with decreasing SZA than the detection limit does. Similar ranges (around SZA of 90°) are used in previous studies e.g. by Kühl et al. 2004b and Hommel et al., 2014. We agree that it would be interesting to investigate also lower SZAs (especially given the better performance of TROPOMI) but we have limited this study to this one SZA range to keep the study in limits.

We added this information to the text of the manuscript (before L120):

"OClO SCDs for SZAs between 89 and 90° during different winters are analysed. This SZA range is motivated by a larger ratio between the OClO SCDs and the detection limit in this range, i.e. for smaller SZA the amplitude of the observed OClO SCDs decreases faster with decreasing SZA than the detection limit does. Similar ranges (around SZA of 90°) are used in previous studies e.g. by Kühl et al. 2004b and Hommel et al., 2014. Although given the better performance of TROPOMI, it would be possible to investigate also lower SZAs. However, we decided to use only the above mentioned SZA range in order to keep this study in limits."

Page 5, lines 135-137: In order to select meteorological quantities, it is assumed that the retrieved OClO SCDs are mostly sensitive to the 475K potential temperature level, which corresponds roughly to an altitude of 19-20km. How far this assumption is valid? It needs also to be better justified.

Selecting this level we follow earlier studies (Wagner et al., 2001, 2002, Kühl et al., 2004b) where a strong anti-correlation between minimum temperatures and OClO SCDs has been found for this PT level. The altitude corresponds well to the peak of the ozone number density profile at high latitudes (Yang, K. and Liu, X.: Ozone profile climatology for remote sensing retrieval algorithms, Atmos. Meas. Tech., 12, 4745–4778, https://doi.org/10.5194/amt-12-4745-2019, 2019.). At the chosen SZA range (89-90°) the measurements also show a very high sensitivity to the investigated altitudes. We added this information to the manuscript.

Technical corrections:

Page 4, line 91: 'coveradge' -> 'coverage'

Corrected

Some sentences are very long and difficult to follow (e.g. first sentence of Section 3, page 5).

We split the sentence: "In addition, we relate the retrieved OClO SCDs with the Level 2 Polar Stratospheric Cloud provisional version 1.10 product (Pitts et al., 2009). The PSC product, freely provided by (NASA/LARC/SD/ASDC, 2016), is retrieved from the…"

The color bar scale values of the subplot stratospheric $T – T_{NAT}$ (3$^{rd}$ subplot from the top) in figures 4, 7, 10, and 13 are difficult to read.

We modified the figures to eliminate the overlap of the scale values.

---

## Author Comment (AC3)

Comment towards the Editor comment in the access review phase about the motivation to introduce the 'PSC evidence' instead of PSC backscatter ratios.

Besides the motivation provided in the discussion manuscript (i.e. "The advantage of the use of the PSC mask product in our opinion is that it reduces the possibility to misinterpret the aerosol information which would be the case if backscatter data would be used instead. (..) We also consider the detection sensitivity which is provided in the PSC product where the horizontal averaging which was necessary to detect PSC is provided. To be able to match an OClO SCD at a given location which is not altitude resolved with a single piece of information about PSCs, we merge the PSC existence profile information as well as the altitude resolved detection sensitivity to a single generic quantity."), we now investigated in a case study the altitudinal mean of the backscatter ratios and compared them to the PSC evidence as well as to the OClO SCDs. We could not find a benefit of using it as a measure of PSC information. Indeed the PSC evidence showed a slightly better sensitivity towards the OClO SCDs especially for periods with low PSCs where the mean backscatter ratios provide just scatter. We added the study in the Appendix B and added the following information to the main text (end of Sect. 3):

"A sensitivity study we performed (see Appendix B) indicates that the PSC evidence is better suited as an indicator of the presence of PSCs than the mean backscatter ratios, especially for low level PSCs."

---

## Editor Decision (ED1)

**Technical corrections for Manuscript No. ACP-2021-600-Revised**

General: space between number and unit missing in several occasions

P1, L5: Add comma → Here, we compare

P4, L115: add "the" before "meteorological" and "TROPOMI" and I would suggest to split the sentence in two sentences starting the second one with "In Sect. 3……". Otherwise, rephrase the sentence so that it reads more smoothly and it becomes more clear what will be shown/done in these sections.

P4, L122: PSCs → PSC?

P5, L123: Something missing? Add "and thus" so that it reads "and thus consequently plays a role……" or simply add a comma?

P5, L129: write "and" instead of colon between the references

P5, L131: I am not sure if the term "in limits" is correct. I would suggest to write it rather the other way that doing an other/larger SZA range would be beyond the scope of this study.

P5, L143: Abbreviation PT already used, but introduced one line later.

P5, L150: Abbreviation RTM has not been introduced.

P6, L156: PT already introduced.

P7, L188: boolean → "a" boolean (?)

P8, L215: add comma after "plotted"

P11, L228: write "degree" or use the degree sign instead of "deg"

P14, L247: space after full stop missing

P14, L249: Add  comma after "However"

P16, L252: I would suggest to rephrase as follows: " …...complete lack of NAT and STS PSCs". Further, what exactly do you mean with here? At a certain day,  for this winter or this time period? This should be more clearly stated than solely "here".

P16, L255: very → quite

P16, L260: barelly →  barely

P16, L260: rephrase as follows: "at the longitudes (around 120°W) at which the largest OclO SCDs are observed".

P17, L286: for a few → at a few   and add "only? So that it reads "only at a few"

P17, L289: From the other hand → On the other hand

P17, L294: I would suggest to make a new sentence: "Further, these look like remnants of earlier chlorine activation"

P18,  297: add either a comma after T_NAT or continue with "and"

P18, L301: very → quite

P18, L302: skip "very"

P18, L306: of the → for the

P18, L316: drop T_NAT → drop below T_NAT

P18, L318: till → until

P18, L323: change as follows: …...appear to be mixed with air from outside the polar vortex (with low PV values)

P18, L323: What is "8" the PV Value or date?

P19, L337: Sentence not clear → check. Maximum of what?

P21, L338: rather "could" than "can"?

P21, L345: space between opening parenthesis and number obsolete

P22, L350-351: Check sentence. Something missing here?

P22, L252: Same here? "but changes during the season" feels a bit lost.

P22, L355: lowers → decrease

P22, L358: "in the SH" obsolete → remove

P24, L364: beginning October → beginning of October

P24, L389: add comma after "seen"?

P27, L390: sudden warming →sudden stratospheric warming

P28, L396: SH winter 2020 → the SH winter 2020

P28, L398: I guess this holds for the time period October to December, thus I would suggest to write that more clearly: " during the time period October to December"

P28, L401: either it should read "In June and August" or "During June to August"

P28, L410: of the → within?

P29, L433: Here it is not clear if you mean with the number the vortex size or the OClO column. Thus, you should clearly state here that these are the OClO values.

P29, L445: add comma after "both" and "horizontally" ?

P30, L465: tipically → typically

P30, L469: righ → right

P30, L474: Add comma after "Therefore"

P31, L484: voundaries → boundaries

Fig A4: x-labels are not appearing fully

P34, L490: Add comma after "Here"

P34, L491: Add comma after "Therefore"

P34, L493: space between comma and "and" missing

P34, L495: right for just May → right just for May (and better to write "only" or "solely" instead of "just"?)

P34, L496: axis scales → y-axis scales

P34, L503: plot correlation → plot the correlation

P34, L504: respectivelly → respectively

P34, L504: I would suggest to write it as follows:"…..between the OClO SCDs and either the PSC evidence or the mean backscatter ratio…..."

P35, L507: datapoints → data points

---

## Author Response (AR2)

We thank the editor very much for the very helpful suggestions.

Technical corrections for Manuscript No. ACP-2021-600-Revised

General: space between number and unit missing in several occasions

We correct as suggested

P1, L5: Add comma → Here, we compare

We correct as suggested

P4, L115: add "the" before "meteorological" and "TROPOMI" and I would suggest to split the sentence in two sentences starting the second one with "In Sect. 3......". Otherwise, rephrase the sentence so that it reads more smoothly and it becomes more clear what will be shown/done in these sections.

We split as suggested and rephrase:

The article is structured as follows: in Sect.~\ref{sect:time} the methodology for comparing the meteorological parameters and the TROPOMI OClO SCDs  are introduced. In Sect.~\ref{sect:pscs} the methodology for comparison of the TROPOMI OClO SCDs with CALIPSO PSCs dataset are described.

P4, L122: PSCs → PSC?

In abstract and on other before we have used "CALIOP PSC observations", we make this consistent throughout the paper

P5, L123: Something missing? Add "and thus" so that it reads "and thus consequently plays a role......" or simply add a comma?

We add a comma

P5, L129: write "and" instead of colon between the references

Done

P5, L131: I am not sure if the term "in limits" is correct. I would suggest to write it rather the other way that doing an other/larger SZA range would be beyond the scope of this study.

We follow the suggestion by reformulating:

Such an investigation, however, is beyond the scope of this study.

P5, L143: Abbreviation PT already used, but introduced one line later.

We fix this

P5, L150: Abbreviation RTM has not been introduced.

We write it out: radiative transfer modelling

P6, L156: PT already introduced.

Fixed

P7, L188: boolean → "a" boolean (?)

Yes, a boolean, We add "a"

P8, L215: add comma after "plotted"

Done

P11, L228: write "degree" or use the degree sign instead of "deg"

We correct by using degree sign

P14, L247: space after full stop missing

Corrected

P14, L249: Add comma after "However"

Corrected

P16, L252: I would suggest to rephrase as follows: " ......complete lack of NAT and STS PSCs".
Further, what exactly do you mean with here? At a certain day, for this winter or this time period?
This should be more clearly stated than solely "here".

We replace "here" with "this time period" since it refers to the beginning of February 2018 as mentioned at the beginning
of the paragraph.

P16, L255: very → quite

Done

P16, L260: barelly → barely

Corrected

P16, L260: rephrase as follows: "at the longitudes (around 120°W) at which the largest OclO SCDs
are observed".

Done

P17, L286: for a few → at a few and add "only? So that it reads "only at a few"

Done

P17, L289: From the other hand → On the other hand

Done

P17, L294: I would suggest to make a new sentence: "Further, these look like remnants of earlier chlorine activation"

We follow the suggestion

P18, 297: add either a comma after T_NAT or continue with "and"

We add a comma

P18, L301: very → quite

Done

P18, L302: skip "very"

Done

P18, L306: of the → for the

Done

P18, L316: drop T_NAT → drop below T_NAT

Done

P18, L318: till → until

Done

P18, L323: change as follows: ......appear to be mixed with air from outside the polar vortex (with low PV values)

Changed

P18, L323: What is "8" the PV Value or date?

It is a typo. There is a reference to Fig. 8, so we add Fig. before 8.

P19, L337: Sentence not clear → check. Maximum of what?

There should be "if" instead of "and", so it reads "maximum of OClO SCDs". We correct.

P21, L338: rather "could" than "can"?

We modify as suggested

P21, L345: space between opening parenthesis and number obsolete

Fixed

P22, L350-351: Check sentence. Something missing here?

Second part of the sentence "also enhanced PSC evidences are found" is also redundant as it tries to say the same as the rest, so we skip it.

P22, L252: Same here? "but changes during the season" feels a bit lost.

We skip "but" to make the sentence more clear

P22, L355: lowers → decrease

We change as suggested

P22, L358: "in the SH" obsolete → remove

Done

P24, L364: beginning October → beginning of October

Done

P24, L389: add comma after "seen"?

Done

P27, L390: sudden warming →sudden stratospheric warming

Done

P28, L396: SH winter 2020 → the SH winter 2020

Done

P28, L398: I guess this holds for the time period October to December, thus I would suggest to write that more clearly: " during the time period October to December"

Done

P28, L401: either it should read "In June and August" or "During June to August"

It should read "During June to August". We modify.

P28, L410: of the → within?

We follow the suggestion

P29, L433: Here it is not clear if you mean with the number the vortex size or the OClO column.
Thus, you should clearly state here that these are the OClO values.

We move "reaching at maximum 3 ×1014 cm−2" to the beginning of the sentence after "OClO SCDs", to make this clear.

P29, L445: add comma after "both" and "horizontally" ?

If the comma should or could be used then before "both" and after "horizontally". We would leave it up to the English proofreading service.

P30, L465: tipically → typically

Corrected

P30, L469: righ → right

Corrected

P30, L474: Add comma after "Therefore"

Added

P31, L484: voundaries → boundaries

Fixed

Fig A4: x-labels are not appearing fully

Fixed

P34, L490: Add comma after "Here"

Added

P34, L491: Add comma after "Therefore"

Added

P34, L493: space between comma and "and" missing

Added

P34, L495: right for just May → right just for May (and better to write "only" or "solely" instead of "just"?)

Corrected

P34, L496: axis scales → y-axis scales

The scales are different for both x- and y- axis, so we write change to "x- and y-axis scales"

P34, L503: plot correlation → plot the correlation

Corrected

P34, L504: respectivelly → respectively

Corrected

P34, L504: I would suggest to write it as follows:".....between the OCIO SCDs and either the PSC evidence or the mean backscatter ratio......"

Rewritten as suggested

P35, L507: datapoints → data points

Corrected